# STRATIFIED GRPO: HANDLING STRUCTURAL HETEROGENEITY IN REINFORCEMENT LEARNING OF LLM SEARCH AGENTS

## ABSTRACT

Large language model (LLM) agents increasingly rely on external tools such as search engines to solve complex, multi-step problems, and reinforcement learning (RL) has become a key paradigm for training them. However, the trajectories of search agents are structurally heterogeneous, where variations in the number, placement, and outcomes of search calls lead to fundamentally different answer directions and reward distributions. Standard policy gradient methods, which use a single global baseline, suffer from what we identify and formalize as cross-stratum bias—an "apples-to-oranges" comparison of heterogeneous trajectories. This cross-stratum bias distorts credit assignment and hinders exploration of complex, multi-step search strategies. To address this, we propose Stratified GRPO, whose central component, Stratified Advantage Normalization (SAN), partitions trajectories into homogeneous strata based on their structural properties and computes advantages locally within each stratum. This ensures that trajectories are evaluated only against their true peers. Our analysis proves that SAN eliminates cross-stratum bias, yields conditionally unbiased unit-variance estimates inside each stratum, and retains the global unbiasedness and unit-variance properties enjoyed by standard normalization, resulting in a more pure and scale-stable learning signal. To improve practical stability under finite-sample regimes, we further linearly blend SAN with the global estimator. Extensive experiments on diverse factual QA and deep-research agent benchmarks demonstrate that Stratified GRPO consistently and substantially outperforms GRPO by up to 12.6 points, achieving higher training rewards, greater training stability, and more effective search policies. These results establish stratification as a principled remedy for structural heterogeneity in RL for LLM search agents.

## 1 INTRODUCTION

Large Language Models (LLMs) (Achiam et al., 2023; DeepSeek-AI et al., 2025; Team, 2024) are increasingly being enhanced with external tools, such as search engines, to create powerful agents capable of tackling complex, multi-step tasks (Schick et al., 2023; Yao et al., 2023; Jin et al., 2025; 2024; Trivedi et al., 2023; Asai et al., 2023). Reinforcement learning (RL) has emerged as a powerful paradigm for training these agents, enabling them to learn sophisticated multi-turn reasoning and tool-use strategies directly from outcome-based rewards (Chen et al., 2025; Jin et al., 2025; Song et al., 2025; Zheng et al., 2025).

A key, and often overlooked challenge for applying standard RL to search agents lies in the structural heterogeneity of agent trajectories. Unlike in conventional reinforcement learning with verifiable rewards (RLVR) settings, where trajectories are sampled from the policy model and follow a relatively homogeneous pattern, the trajectories of search agents differ markedly in their structure due to the number, placement, and outcomes of search invocations. For instance, a trajectory with zero search calls is qualitatively different from one with multiple, as the retrieved information can substantially alter subsequent generations and induce distinct behavior modes and, consequently, different reward distributions. Standard policy gradient methods, which often rely on a single global baseline to compute advantages, implicitly assume all trajectories are comparable. This creates a flawed "apples-to-oranges" comparison. In this work, we identify and formalize this fundamental

issue as **cross-stratum bias**. As we show, this structural flaw distorts credit assignment and hinders exploration of complex multi-step search strategies, leading to suboptimal policies.

We propose **Stratified GRPO** to address this fundamental issue. Instead of using a single global baseline, Stratified GRPO uses Stratified Advantage Normalization (SAN), a principled advantage estimator designed for heterogeneous action spaces. The core idea is simple yet powerful: partition trajectories into homogeneous strata, based on their structural properties (e.g., the number of search calls), and then compute advantages within each stratum. By construction, SAN is free from cross-stratum bias, ensuring that each trajectory is evaluated only against its true peers. Our theoretical analysis formalizes the benefits of this approach. We show that SAN removes the cross-stratum bias inherent in global baselines, ensuring fair credit assignment. We further prove that SAN is conditionally unbiased and has unit variance within each stratum, acting as a pure and scale-stable learning signal. Critically, we demonstrate that SAN achieves these superior conditional properties while matching the global unbiasedness and unit variance of standard normalization methods. To enhance practical stability in finite-sample regimes, we further introduce Blended Advantage that robustly combines SAN with the global estimator.

We validate Stratified GRPO through comprehensive experiments on diverse factual QA and deep-research agent benchmarks. The results demonstrate the clear superiority of our method. Stratified GRPO consistently outperforms the standard GRPO baseline, achieving a relative improvement of up to 12.6 points. Furthermore, Stratified GRPO also exhibits higher training rewards, greater training stability, and learns more effective search policies than standard GRPO. These findings provide strong empirical evidence that our principled approach successfully mitigates cross-stratum bias. Our main contributions are as follows:

- We identify and formalize **cross-stratum bias**, a fundamental challenge in policy gradient methods for LLM search agents. We provide a theoretical decomposition that proves this bias arises from using a global baseline across structurally heterogeneous agent trajectories.

- We propose **Stratified GRPO**, a principled RL algorithm that eliminates this cross-stratum bias. Its core component, Stratified Advantage Normalization (SAN), partitions trajectories into homogeneous strata and computes advantages locally, ensuring a fair and stable credit assignment.

- We provide a rigorous theoretical analysis of SAN, proving that it eliminates cross-stratum bias and is conditionally unbiased and has unit variance within each stratum. Crucially, SAN achieves these superior conditional properties while preserving the global unbiasedness and unit variance of standard normalization, yielding a more pure and scale-stable learning signal.

- We demonstrate through extensive experiments on diverse factual QA and deep-research agent benchmarks that Stratified GRPO substantially outperforms GRPO by up to 12.6 points. Our method achieves higher training rewards, improves training stability, and learns more effective search policies.

## 2 RELATED WORK

### 2.1 REINFORCEMENT LEARNING FOR LARGE LANGUAGE MODELS

Reinforcement learning (RL) (Kaelbling et al., 1996; Sutton et al., 1999) has become a central component in post-training large language models (LLMs). The most widely adopted paradigm is Reinforcement Learning from Human Feedback (RLHF), which learns a reward model from human preferences and then optimizes a policy with RL algorithms, most often Proximal Policy Optimization (PPO) (Ouyang et al., 2022; Schulman et al., 2017). Although effective, RLHF can be computationally expensive and brittle due to reward model training and distribution shift. To reduce the cost and instability of explicit reward modeling, direct alignment methods like DPO optimize preference data without training a separate reward model (Rafailov et al., 2023; Zhu et al., 2025). A complementary line of work targets reasoning by exploiting verifiable outcomes using Reinforcement Learning with Verifiable Rewards (RLVR) (DeepSeek-AI et al., 2025; Shao et al., 2024; Ahmadian et al., 2024; Yu et al., 2025). A prominent approach is Group Relative Policy Optimization (GRPO) (Shao et al., 2024), which removes PPO's dependency on a learned value function by using group-based

baselines, and RLOO (Ahmadian et al., 2024) revisits REINFORCE (Williams, 1992) with simplifications tailored to LLM training. However, most of these advances target general conversational capabilities. By contrast, the systematic study of RL algorithms for LLM agents, especially search agents that interleave generation with search over external information and require long-horizon reasoning, remains underexplored. Our work addresses this gap by providing a systematic analysis of the LLM search agent setting and proposing RL algorithms tailored to its unique challenges.

## 2.2 LARGE LANGUAGE MODEL SEARCH AGENTS

Large language models (LLMs) (Achiam et al., 2023; Team, 2024; Zhao et al., 2023) exhibit strong reasoning capabilities (DeepSeek-AI et al., 2025). Building on this capacity, recent work has developed agentic workflows that equip LLMs with external tools for complex problem solving (Schick et al., 2023). A prominent instantiation is the LLM search agent, which treats a search engine as a callable tool at inference time (Schick et al., 2023): the model iteratively proposes queries, retrieves evidence, and updates its reasoning based on retrieved documents (Trivedi et al., 2023). Two main approaches have emerged. One line of work uses carefully designed prompts to instruct LLMs to interleave reasoning and retrieval (Trivedi et al., 2023; Yao et al., 2023). Another line curates trajectories that mix reasoning with search and then applies supervised fine-tuning (Schick et al., 2023; Asai et al., 2023). More recently, several studies (Chen et al., 2025; Jin et al., 2025; Song et al., 2025; Zheng et al., 2025) show that complex search-and-reasoning behaviors can be acquired directly from outcome-based rewards using RL algorithms such as PPO or GRPO. However, these RL applications typically adopt general-purpose algorithms without addressing the specific intricacies of the search agent setting. Our work identifies and formalizes cross-stratum bias as a fundamental challenge for LLM search agents, arising from the heterogeneous nature of agent trajectories that use different search strategies. To overcome this, we propose Stratified GRPO, a principled algorithm designed to eliminate this bias and improve learning more effective search policies for LLM search agents.

## 3 METHODS

We study reinforcement learning (RL) for multi-turn search agents, in which trajectories from different strategies (e.g., varying search counts) are not directly comparable. We show that the structural heterogeneity of trajectories induces a **cross-stratum bias** whenever advantages are computed with baselines that ignore the heterogeneity driver. We then introduce **Stratified Advantage Normalization (SAN)**, an estimator that partitions trajectories into homogeneous strata and normalizes advantages therein, and analyze its statistical and structural properties.

### 3.1 RL FOR MULTI-TURN SEARCH AGENTS

Following Jin et al. (2025), we formulate the task of training a multi-turn search agent as a reinforcement learning problem. The agent, parameterized by a policy $p_\theta$, interacts with a search engine by interleaving token generation with search queries. For a given prompt $x \sim \mathcal{D}$, the agent generates a trajectory $\tau \sim p_\theta(\cdot \mid x)$, which consists of a sequence of actions of generating tokens or issuing searches. Upon completion, the trajectory receives a scalar reward $R(\tau)$ that reflects the quality of the final response. The objective is to maximize the expected reward:

$$\max_\theta J(\theta) = \mathbb{E}_{x\sim\mathcal{D},\tau\sim p_\theta(\tau|x)}[R(\tau)]. \tag{1}$$

This problem is usually optimized using policy gradient methods.

### 3.2 CROSS-STRATUM BIAS IN POLICY-GRADIENT BASELINES

In search agents, trajectory heterogeneity is significant: the number, content, and outcomes of tool calls vary, resulting in strata with systematically different answer directions and reward distributions. A global baseline is poorly suited because it implicitly assumes that all strategies are comparable. Using a global baseline across this heterogeneous mixture induces a cross-stratum bias, forcing an "apples-to-oranges" comparison. This flaw can be formalized by decomposing the global advantage.

**Notation.** Let $B = \{\tau_1, \ldots, \tau_K\}$ denote a batch of $K$ trajectories sampled i.i.d. from $p_\theta$ for a fixed prompt $x$. The batch is partitioned into $I$ non-empty strata $B_0, \ldots, B_{I-1}$ according to a predefined structure (e.g., search count), with $|B_k| = n_k$. Each trajectory $\tau_i$ has reward $R_i$. Let $\bar{R}_{\text{global}} = \frac{1}{K} \sum_{j=1}^{K} R(\tau_j)$ be the global mean reward of the batch, and define the stratum-specific mean $\bar{R}_k = \frac{1}{n_k} \sum_{\tau_i \in B_k} R_i$. This gives rise to two natural advantage estimators:

$$\hat{A}_G(\tau_i) = R_i - \bar{R}_{\text{global}} \quad \text{(global)}, \qquad \hat{A}_S(\tau_i) = R_i - \bar{R}_k \quad \text{(stratified, } \tau_i \in B_k \text{)}.$$

**Advantage Bias and Variance Decomposition.** The following result shows how the global advantage differs systematically from the stratified advantage.

**Proposition 1** (Advantage Decomposition). *For any trajectory $\tau_i \in B_k$, the global advantage decomposes as*

$$\hat{A}_G(\tau_i) = \hat{A}_S(\tau_i) + \underbrace{(\bar{R}_k - \bar{R}_{\text{global}})}_{\text{cross-stratum bias}}.$$

*(The proof follows directly from the definitions).* This decomposition highlights a **structural flaw: the cross-stratum advantage bias** is a deterministic offset applied uniformly within each stratum, unfairly penalizing trajectories from low-reward strata while favoring those from high-reward strata. Importantly, this offset is **precisely the source of the excess variance of** $\hat{A}_G$.

**Theorem 1** (Variance Reduction via Stratified Baselines). *The empirical variances of the stratified and global advantage estimators satisfy $\text{Var}[\hat{A}_S] \leq \text{Var}[\hat{A}_G]$. Moreover, the reduction is exactly the variance induced by the cross-stratum bias:*

$$\text{Var}[\hat{A}_G] - \text{Var}[\hat{A}_S] = \frac{1}{K} \sum_{k=0}^{I-1} n_k \left( \bar{R}_k - \bar{R}_{\text{global}} \right)^2.$$

*Equality holds if and only if all stratum means coincide, i.e. $\bar{R}_0 = \bar{R}_1 = \cdots = \bar{R}_{I-1}$. Otherwise, stratification strictly reduces variance.*

*(Proof in Appendix A.1).* Together, Proposition 1 and Theorem 1 formalize the flaw of a global baseline: the cross-stratum bias is precisely the term that inflates variance. Stratification corrects this by ensuring comparisons are made only among homogeneous peers, yielding a strictly lower-variance advantage estimator.

The principle in Theorem 1 applies to commonly-used policy gradient methods for LLMs that use a baseline not conditioned on the stratum structure $S$. This includes REINFORCE with a global baseline and RLOO (Ahmadian et al., 2024). For actor-critic methods such as PPO (Schulman et al., 2017), similar effects may occur when the imperfectly learned critic does not condition on $S$, effectively introducing a structural bias similar to that of a global baseline.

### 3.3 STRATIFIED ADVANTAGE NORMALIZATION: DEFINITION AND THEORETICAL GUARANTEES

Building on the principle of stratification, we propose **Stratified Advantage Normalization (SAN)**, an estimator that augments stratification with per-stratum normalization to create a stable, scale-invariant learning signal.

**Definition 1.** *For a given prompt $x$, partition the batch of trajectories into strata $\{B_k(x)\}$ based on a chosen partitioning function (e.g., the search count for search agents). The SAN advantage for a trajectory $\tau_i \in B_k(x)$ is defined as*

$$A_{SAN}(\tau_i) = \frac{R(\tau_i) - \widehat{\mu}_k(x)}{\widehat{\sigma}_k(x) + \varepsilon}, \tag{2}$$

*where $\widehat{\mu}_k(x)$ and $\widehat{\sigma}_k(x)$ are the empirical mean and standard deviation of rewards in stratum $B_k(x)$, $\varepsilon > 0$ is a small constant for numerical stability.*

**Advantage Invariance and Robustness.** A key property of SAN is its robustness to the scale and shift of rewards, a desirable feature formalized below.

**Proposition 2** (Invariance to Positive Affine Reward Transforms). *Suppose $\varepsilon = 0$. The SAN advantage $A_{SAN}(\tau)$ is invariant under any positive affine transformation of the rewards, $R'(\tau) = aR(\tau)+b$ with $a > 0$. That is, $A'_{SAN}(\tau) = A_{SAN}(\tau)$.*

*(Proof in Appendix A.2).* The invariance shown in Proposition 2 makes SAN robust to arbitrary changes in reward scaling. While a small $\varepsilon > 0$ slightly breaks this perfect invariance in practice, it ensures numerical stability.

**Variance Decomposition.** In Theorem 1, we analyzed the variance reduction of the stratified estimator, $\hat{A}_S$, relative to the global estimator, $\hat{A}_G$. We now extend this comparison to the fully stratified and normalized advantage, $A_{\text{SAN}}$, providing an exact decomposition of the variance difference.

**Theorem 2** (Variance Decomposition for Normalized Stratified Advantage). *Let $A_{\text{SAN}}(\tau_i)$ be the stratified and normalized advantage. It is related to the global one $\hat{A}_G$ by the following exact decomposition:*

$$
\text{Var}[\hat{A}_G] - \text{Var}[A_{\text{SAN}}] = \underbrace{\frac{1}{K} \sum_{k=0}^{I-1} n_k (\bar{R}_k - \bar{R}_{\text{global}})^2}_{\textit{Term A: Between-Stratum Variance}} + \underbrace{\frac{1}{K} \sum_{k=0}^{I-1} n_k \hat{\sigma}_k^2 \left( 1 - \frac{1}{(\hat{\sigma}_k + \varepsilon)^2} \right)}_{\textit{Term B: Normalization Effect}}.
$$

*(Proof in Appendix A.3).* Theorem 2 formalizes how SAN improves over the global baseline. Term A quantifies the between-stratum variance that arises from heterogeneous stratum means: by centering rewards within each stratum, SAN fully removes this structural bias (Proposition 1), ensuring that gradient estimates are not artificially inflated by cross-stratum differences. Term B captures the effect of per-stratum normalization. While it can be positive or negative, it primarily stabilizes the scale of rewards within each stratum, producing a more consistent and numerically robust learning signal. These finite-sample insights will be complemented by the large-sample conditional properties in Theorem 4.

**The Gradient Bias Trade-off.** While SAN eliminates the structural bias of a global baseline, its expected gradient admits a particularly clean form. Specifically, it decomposes into a weighted sum of the true within-stratum gradients, with weights determined by the stratum probabilities and scaled by their inverse standard deviations. The following theorem formalizes this result.

**Theorem 3** (Population SAN Expectation). *Let $S = S(\tau; \theta)$ be a discrete stratum assignment that may depend on $\theta$, and define the per-trajectory SAN advantage*

$$
A_{\text{SAN}}(\tau) := \frac{R(\tau) - \mu_S(\theta)}{\sigma_S(\theta) + \varepsilon},
$$

*where $\mu_k(\theta) = \mathbb{E}_\theta[R \mid S = k]$ and $\sigma_k(\theta) > 0$ are the stratum-wise mean and standard deviation, and $\varepsilon > 0$ is a small regularizer. Then, under standard regularity conditions allowing differentiation under the expectation,*

$$
\mathbb{E}_\theta\big[A_{\text{SAN}}(\tau)\,\nabla_\theta \log p_\theta(\tau)\big] = \sum_k \frac{p_k(\theta)}{\sigma_k(\theta) + \varepsilon}\,\nabla_\theta \mu_k(\theta), \quad p_k(\theta) := \Pr_\theta(S = k), \tag{3}
$$

*i.e., the population SAN estimator exactly targets the weighted sum of within-stratum gradients, even when the strata depend on $\theta$.*

*(Proof in Appendix A.4).* Given stratum $k$, $\mu_k(\theta) := \mathbb{E}_\theta[R(\tau) \mid S = k]$. Its gradient with respect to the policy parameters $\theta$ is $\nabla_\theta \mu_k(\theta) = \nabla_\theta \mathbb{E}_\theta[R(\tau) \mid S = k]$. Applying the *score function identity* (or likelihood ratio trick) conditional on $S = k$ gives:

$$
\nabla_\theta \mu_k(\theta) = \nabla_\theta \sum_\tau R(\tau)\, p_\theta(\tau \mid S = k) = \sum_\tau R(\tau)\, \nabla_\theta p_\theta(\tau \mid S = k)
$$

$$
= \sum_\tau R(\tau)\, p_\theta(\tau \mid S = k)\, \nabla_\theta \log p_\theta(\tau \mid S = k)
$$

$$
= \mathbb{E}_\theta\big[R(\tau)\, \nabla_\theta \log p_\theta(\tau \mid S = k) \,\big|\, S = k\big].
$$

Subtracting the conditional mean inside the expectation does not change the result, because $\mathbb{E}_\theta[\nabla_\theta \log p_\theta(\tau \mid S = k) \mid S = k] = 0$. Thus, we can write

$$\nabla_\theta \mu_k(\theta) = \mathbb{E}_\theta\big[(R(\tau) - \mu_k(\theta))\,\nabla_\theta \log p_\theta(\tau \mid S = k)\,\big|\, S = k\big].$$

This shows explicitly that $\nabla_\theta \mu_k(\theta)$ is exactly the *true per-stratum policy gradient*. Consequently, **the SAN gradient estimator, which combines these per-stratum gradients weighted by** $p_k(\theta)/(\sigma_k(\theta)+\varepsilon)$**, targets a** *weighted sum of the true per-stratum gradients***, even when strata are** $\theta$**-dependent.**

**Structural Superiority over Global Normalization.**     To fully appreciate the principled design of SAN, we briefly contrast it with the standard Global Normalization (GN) used in GRPO. While GN standardizes rewards globally, i.e., $A_{GN} = (R - \bar{R}_{global})/\hat{\sigma}_{global}$, it ignores the structural heterogeneity of the search trajectories. We formally analyze this difference in Appendix B. Our analysis reveals two critical flaws in GN:

1. **Cross-Stratum Offset Bias:** We prove that the GN advantage decomposes into a rescaled SAN advantage plus a *cross-stratum offset* term $\Delta_k$, explicitly satisfying (*cf.* Proposition 3 in Appendix B):
$$A_{GN} = \alpha_k A_{SAN} + \Delta_k, \tag{4}$$
   where $\alpha_k = \frac{\hat{\sigma}_k + \epsilon}{\hat{\sigma}_{global} + \epsilon}$ accounts for the inconsistent scaling, and $\Delta_k = \frac{\hat{\mu}_k - \bar{R}_{global}}{\hat{\sigma}_{global} + \epsilon}$ represents the systematic mean shift. This offset $\Delta_k$ systematically penalizes trajectories in strata with lower mean rewards (e.g., those attempting complex searches that may initially fail), thereby suppressing exploration of diverse search strategies. This explains the empirical observation (Figure 1) where standard GRPO struggles to explore beyond single-step search.

2. **Inconsistent Conditional Signal:** While both estimators appear globally unbiased, strictly within each stratum, their behaviors diverge significantly. As formalized below, SAN acts as a pure signal carrier, whereas GN is conditionally biased and inconsistently scaled:

**Theorem 4** (Conditional Properties of SAN and GN). *Let $\varepsilon = 0$. Let $\mu_k(x), \sigma_k^2(x) \neq 0$ be the reward mean and variance for stratum $k$, and $\mu(x), \sigma^2(x) \neq 0$ be the global statistics. In the large-sample limit, the conditional properties for any stratum $k$ are:*

> ***Bias:***   $\mathbb{E}[A_{\text{SAN}} \mid k, x] = 0,$       $\mathbb{E}[A_{\text{GN}} \mid k, x] = \dfrac{\mu_k(x) - \mu(x)}{\sigma(x)}.$

> ***Variance:***   $\text{Var}(A_{\text{SAN}} \mid k, x) = 1,$       $\text{Var}(A_{\text{GN}} \mid k, x) = \dfrac{\sigma_k^2(x)}{\sigma^2(x)}.$

Consequently, SAN provides a *pure* (zero-mean) and *scale-stable* (unit-variance) learning signal within every stratum. In contrast, GN introduces a systematic bias proportional to the cross-stratum mean difference and a variance that fluctuates based on reward heterogeneity. By eliminating this structural bias, SAN ensures that credit assignment depends solely on the relative quality of a trajectory among its structural peers, rather than global statistics.

Despite these stark conditional differences, the normalization step effectively masks them at the global level. The following theorem establishes that both estimators are mathematically constrained to have identical global moments:

**Theorem 5** (Global Moments of SAN and GN). *Under the notations of Theorem 4, let $\varepsilon = 0$. The large-sample (population) advantages satisfy:*

> ***(a) Global Means:***   $\mathbb{E}[A_{\text{SAN}} \mid x] = 0,$   $\mathbb{E}[A_{\text{GN}} \mid x] = 0.$
> ***(b) Global Variances:***   $\text{Var}(A_{\text{SAN}} \mid x) = \text{Var}(A_{\text{GN}} \mid x) = 1.$

This global equivalence highlights why macroscopic metrics (like global variance) fail to capture the structural flaws of GN, further justifying our focus on conditional properties and gradient stability.

In summary, Stratified Advantage Normalization (SAN) applies per-stratum standardization (Equation (2)), making advantages robust to reward scaling (Proposition 2). The population characterization (Theorem 3) ensures it exactly targets a weighted sum of true per-stratum gradients.

Conditional-moment analysis (Theorem 4) further demonstrates that SAN provides a peer-to-peer learning signal—each trajectory is evaluated relative to others in the same stratum—unlike Global Normalization, which mixes heterogeneous behaviors and introduces conditional bias. Together, these results establish SAN as a principled, stable, and peer-consistent estimator under heterogeneous trajectory distributions.

## 4 Blended Advantage for Finite-Sample Stability

From Theorem 4, SAN yields a *pure, scale-stable* learning signal: it is conditionally unbiased within each stratum and has unit conditional variance. In contrast, while GN reintroduces a cross-stratum offset and inconsistent scaling, it inherently couples information across strata via the global offset term $\Delta_k$, as shown in the decomposition in Equation (4).

Nevertheless, in the finite sample regime, SAN may face a practical challenge when some strata contain very few trajectories, which may cause noisy advantage estimates. Specifically, the empirical statistics $(\hat{\mu}_k, \hat{\sigma}_k)$ become noisy, and the "pure" SAN estimator might become unstable. To address this issue, we employ a linear combination of SAN and GN that preserves SAN's local purity while borrowing GN's global signal to stabilize small strata.

**Definition 2** (Blended Advantage). *For $\tau \in B_k(x)$, define*

$$A_{\text{blend}}(\tau) \ = \ \alpha\, A_{\text{SAN}}(\tau) \ + \ (1-\alpha)\, A_{\text{GN}}(\tau), \qquad \alpha \in [0,1]. \tag{5}$$

The endpoints recover known estimators: $\alpha = 1$ yields SAN, and $\alpha = 0$ yields GN. Incorporating the blended advantage into SAN gives our practical method, *Stratified GRPO* (Algorithm 1).

## 5 Experiments

We evaluate Stratified GRPO factual QA tasks following Search-R1 (Jin et al., 2025) and further evaluate its effectiveness on challenging deep-research agent tasks (Jiang et al., 2025). Further analysis reveals that compared to standard GRPO, our method also achieves superior training stability and learns a more effective multi-step search policy.

### 5.1 Experimental Setup

**Models and Training.** For factual QA tasks, we conduct experiments on the Qwen-2.5-3B Base and Instruct models (Yang et al., 2024). For retrieval, we use the 2018 Wikipedia dump (Karpukhin et al., 2020) as the knowledge source and E5 (Wang et al., 2022) as the retriever, fetching the top-3 passages per query. Following the setup in Jin et al. (2025), we construct our training set by merging the training splits of Natural Questions (NQ) (Kwiatkowski et al., 2019) and HotpotQA (Yang et al., 2018). We use Exact Match (EM) as the training reward. For deep-research agent tasks, we follow the same settings in Jiang et al. (2025). Specifically, we use Qwen3-8B (Yang et al., 2025) as the base model and equip it with a Google search engine and a sandboxed Python executor as tools. Additional experiment details are in Appendix E.

**Stratification Heuristic.** In all experiments, we stratify trajectories within each prompt based on the number of tool calls. This design is motivated by our analysis of heterogeneity drivers in Section 3.2, which identifies the number of tool calls as the primary factor distinguishing trajectory structures and reward distributions. This stratification heuristic ensures that the advantage is computed among structurally homogeneous peers.

**Evaluation Benchmarks.** For factual QA tasks, we evaluate performance on seven diverse question-answering datasets. These include three single-hop QA benchmarks: NQ (Kwiatkowski et al., 2019), TriviaQA (Joshi et al., 2017), and PopQA (Mallen et al., 2023); and four multi-hop QA benchmarks: HotpotQA (Yang et al., 2018), 2WikiMultiHopQA (Ho et al., 2020), MuSiQue (Trivedi et al., 2022), and Bamboogle (Press et al., 2023). Consistent with standard practice (Yu et al., 2024; Jin et al., 2025), EM is used as the evaluation metric. For deep-research agent tasks, we evaluate on the General AI Assistant (GAIA) benchmark (Mialon et al., 2023).

Table 1: Experiment results on seven QA benchmarks. **Bold** denotes best results.

| Methods | Single-Hop QA | | | Multi-Hop QA | | | | Avg. |
|---|---|---|---|---|---|---|---|---|
| | NQ | TriviaQA | PopQA | HotpotQA | 2Wiki | Musique | Bamboogle | |
| *Non-RL Baselines* | | | | | | | | |
| Direct Generation | 10.6 | 28.8 | 10.8 | 14.9 | 24.4 | 2.0 | 2.4 | 13.4 |
| SFT | 24.9 | 29.2 | 10.4 | 18.6 | 24.8 | 4.4 | 11.2 | 17.6 |
| RAG | 34.8 | 54.4 | 38.7 | 25.5 | 22.6 | 4.7 | 8.0 | 27.0 |
| Search-o1 | 23.8 | 47.2 | 26.2 | 22.1 | 21.8 | 5.4 | 32.0 | 25.5 |
| IRCoT | 11.1 | 31.2 | 20.0 | 16.4 | 17.1 | 6.7 | 24.0 | 18.1 |
| *Qwen2.5-3B-Base* | | | | | | | | |
| Search-R1 | 40.6 | 58.7 | 43.5 | 28.4 | 27.3 | 4.9 | 8.8 | 30.3 |
| R1 | 22.6 | 45.5 | 17.3 | 20.1 | 26.8 | 5.5 | 22.4 | 22.9 |
| ReSearch | 42.7 | 59.7 | 43.0 | 30.5 | 27.2 | 7.4 | 12.8 | 31.9 |
| GRPO | 45.2 | 61.2 | **43.8** | 32.6 | 29.7 | 7.8 | 12.9 | 33.3 |
| Stratified GRPO | **45.9** | **61.4** | 43.0 | **40.8** | **39.9** | **17.7** | **42.7** | **41.6** |
| *Qwen2.5-3B-Instruct* | | | | | | | | |
| Search-R1 | 34.1 | 54.5 | 37.8 | 32.4 | 31.9 | 10.3 | 26.4 | 32.5 |
| R1 | 21.0 | 44.9 | 17.1 | 20.8 | 27.5 | 6.0 | 19.2 | 22.4 |
| ReSearch | 36.5 | 57.1 | 39.5 | 35.1 | 27.2 | 9.5 | 26.6 | 33.1 |
| GRPO | 33.4 | 52.9 | 36.7 | 26.5 | 27.4 | 6.4 | 21.0 | 29.2 |
| Stratified GRPO | **44.5** | **60.9** | **44.3** | **41.0** | **37.3** | **16.9** | **38.7** | **40.5** |

**Baselines.** For factual QA tasks, we compare Stratified GRPO against a comprehensive set of non-RL and RL methods. Non-RL methods include Direct Generation, Supervised Fine-Tuning (SFT), RAG (Lewis et al., 2020), Search-o1 (Li et al., 2025a), and IRCoT (Trivedi et al., 2023). RL Methods includes Search-R1 (Jin et al., 2025), RL without search (R1) (DeepSeek-AI et al., 2025), ReSearch (Chen et al., 2025), and GRPO (Shao et al., 2024). Most baseline results are cited from Jin et al. (2025) since their experiment setting is consistent with ours. For deep-research agent tasks, we compare Stratified GRPO against large models without tools, agents with tool integrated reasoning, and GRPO. For large models without tools, we include Qwen3-32B-thinking (Yang et al., 2025), DeepSeek-R1-32B, DeepSeek-R1-671B (DeepSeek-AI et al., 2025), QwQ-32B (Team, 2025), and GPT-4o (Achiam et al., 2023) as baselines. For agents with tool integrated reasoning, we include RAG, Search-o1, WebThinker (Li et al., 2025b), and ReAct (Yao et al., 2023) as baselines.

## 5.2 MAIN RESULTS

**Results on Factual QA Tasks.** The factual QA experiment results, summarized in Table 1, demonstrate that Stratified GRPO consistently outperforms all baseline methods across seven QA benchmarks. On average, our method improves upon GRPO by up to 11.3 points and surpasses the best-performing baseline by up to 8.3 points. The advantage is particularly pronounced on multi-hop benchmarks, where Stratified GRPO achieves an average performance gain of up to 14.5 points over the strongest baseline. We attribute this success to our method's ability to eliminate systematic cross-stratum bias, enabling more effective learning from trajectories with varying search counts.

Notably, the consistent outperformance of Stratified GRPO over the PPO-based Search-R1 empirically suggests that the challenge of cross-stratum bias should also hold true for PPO. While our analysis focuses on the explicit bias in policy gradient's global baseline, this issue likely manifests in algorithms like PPO through the difficulty of training an accurate value function for structurally diverse trajectories. Our results, therefore, indicate that a principled handling of trajectory structure is a key factor in robustly training effective search agents.

**Results on Deep-Research Agent Tasks.** To further evaluate the effectiveness of Stratified GRPO on real-world multi-step tool-use scenarios, we compare its performance on GAIA benchmark, a rigorous suite designed to test comprehensive capabilities in reasoning, web browsing, and tool proficiency, against a comprehensive set of strong large models and agents.

Table 2 highlights the substantial gains achieved by Stratified GRPO, which outperforms the standard GRPO baseline by an average of **12.6 points**. Notably, this performance gap widens on the

Table 2: Performance on the GAIA benchmark. Stratified GRPO significantly outperforms baselines, particularly on complex tasks (Level 2 & 3), demonstrating superior performance in challenging real-world tool-use scenarios. **Bold** denotes best results.

| Method | Level 1 | Level 2 | Level 3 | Avg. |
|---|---|---|---|---|
| *Reasoning without Tool* | | | | |
| Qwen3-32B-thinking | 26.2 | 12.1 | 0.0 | 14.9 |
| DeepSeek-R1-32B | 21.5 | 13.6 | 0.0 | 14.2 |
| QwQ-32B | 30.9 | 6.5 | 5.2 | 18.9 |
| GPT-4o | 23.1 | 15.4 | 8.3 | 17.5 |
| DeepSeek-R1-671B | 40.5 | 21.2 | 5.2 | 25.2 |
| *Tool Integrated Reasoning (Qwen3-8B)* | | | | |
| Vanilla RAG | 28.2 | 15.4 | 16.7 | 20.4 |
| Search-o1 | 35.9 | 15.4 | 0.0 | 21.4 |
| WebThinker | 43.6 | 11.5 | 0.0 | 22.3 |
| ReAct | 35.9 | 17.3 | 8.3 | 23.3 |
| *RL-based Method* | | | | |
| Qwen3-8B | 28.1 | 15.4 | 16.7 | 20.4 |
| + GRPO | 48.7 | 32.7 | 16.7 | 36.9 |
| **+ Stratified GRPO** | **61.5** | **44.2** | **33.3** | **49.5** |

Table 3: Ablation study analyzing the components of Stratified GRPO, comparing the baseline GRPO, GRPO with SAN, and our full Stratified GRPO.

| Model Variants | Single-Hop QA | | | Multi-Hop QA | | | | |
|---|---|---|---|---|---|---|---|---|
| | NQ | TriviaQA | PopQA | HotpotQA | 2wiki | Musique | Bamboogle | Avg. |
| *Qwen2.5-3B-Base* | | | | | | | | |
| GRPO | 45.2 | 61.2 | **43.8** | 32.6 | 29.7 | 7.8 | 12.9 | 33.3 |
| w/ SAN | 43.7 | 59.3 | 41.1 | 36.6 | 38.4 | 12.6 | 25.0 | 36.7 |
| Stratified GRPO | 45.9 | **61.4** | 43.0 | **40.8** | **39.9** | **17.7** | **42.7** | **41.6** |
| *Qwen2.5-3B-Instruct* | | | | | | | | |
| GRPO | 33.4 | 52.9 | 36.7 | 26.5 | 27.4 | 6.4 | 21.0 | 29.2 |
| w/ SAN | 42.5 | 60.1 | 44.2 | 39.4 | **41.0** | 16.0 | 36.3 | 39.9 |
| Stratified GRPO | **44.5** | **60.9** | **44.3** | **41.0** | 37.3 | **16.9** | **38.7** | **40.5** |

most challenging tasks (Level 3), where Stratified GRPO yields a **2× relative improvement** over GRPO (33.3 vs. 16.7). These results confirm the domain-agnostic nature of our approach: the benefits of mitigating cross-stratum bias extend effectively to complex agents involving open-web search, code execution, and long-horizon planning.

## 5.3 ANALYSIS

In this section, we conduct an ablation study to isolate the contributions of Stratified GRPO's components. We also empirically analyze its training dynamics in comparison to GRPO. We include further analysis of our Stratified GRPO in Section D

**Ablation Studies.** We perform an ablation study to analyze the contribution of each component of our proposed method. As shown in Table 3, each component provides a clear benefit. SAN alone yields significant gains over the baseline GRPO. The subsequent addition of advantage blending further enhances performance, establishing the effectiveness of the full Stratified GRPO algorithm, which consistently outperforms the other variants, especially on complex multi-hop QA tasks.

**Improved Reward and Training Stability.** Figure 1 (left) illustrates the training reward curves. For the base model, Stratified GRPO consistently achieves higher rewards than the standard GRPO baseline. More notably, when applied to the instruct model, standard GRPO suffers from training collapse—a known instability issue also documented by prior works (Jin et al., 2025). In contrast, Stratified GRPO maintains a stable and monotonically increasing reward signal, demonstrating its superior stability and learning efficiency.

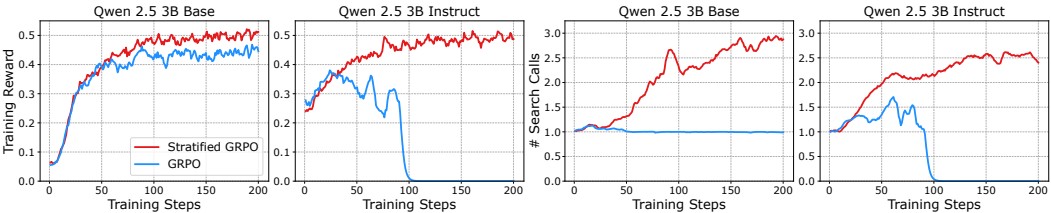

Figure 1: Training dynamics of Stratified GRPO and GRPO. The left plots show training rewards, and the right plots show the number of search calls per question over training steps.

Table 4: Sensitivity analysis of the blending coefficient $\alpha$ on Qwen-2.5-3B-Base. The method shows robust improvements over a wide range of $\alpha$, with the best performance achieved at $\alpha = 0.6$.

| Value of $\alpha$ | NQ | TriviaQA | PopQA | HotpotQA | 2Wiki | Musique | Bamboogle | Avg. |
|---|---|---|---|---|---|---|---|---|
| 0.0 (GRPO) | 45.2 | 61.2 | 43.8 | 32.6 | 29.7 | 7.8 | 12.9 | 33.3 |
| 0.2 | 45.1 | 61.8 | 43.1 | 32.2 | 29.5 | 7.6 | 14.5 | 33.4 |
| 0.4 | 44.6 | 59.8 | 43.2 | 39.6 | 36.5 | 15.6 | 39.5 | 39.8 |
| **0.6 (Default)** | **45.9** | 61.4 | 43.0 | **40.8** | 39.9 | **17.7** | **42.7** | **41.6** |
| 0.8 | 45.1 | 60.1 | **43.9** | 40.3 | **41.9** | 17.5 | 37.9 | 41.0 |
| 1.0 (SAN-only) | 43.7 | 59.3 | 41.1 | 36.6 | 38.4 | 12.6 | 25.0 | 36.7 |

**Learning an Effective Search Policy.** A crucial ability for search agents is learning to identify knowledge gaps and issue search queries accordingly. We analyze this by tracking the average number of search calls per question during training (Figure 1, right). Stratified GRPO successfully learns a policy that converges to approximately 2.5 search calls, indicating it has learned to perform iterative searches. Conversely, the baseline GRPO stagnates at around one search call for the base model and causes training collapse for the instruct model. This is because GRPO's cross-stratum bias prevents it from exploring potentially better search policies. This result demonstrates that Stratified GRPO learns a more effective search policy, which directly translates to its superior performance on multi-hop benchmarks that require sequential information retrieval, as shown in Table 1.

**Robustness of Blending Coefficient $\alpha$.** In Section 4, we introduced the blending coefficient $\alpha$ to balance the structural purity of SAN with the global variance reduction of Group Normalization (GN). To verify the robustness of this hyperparameter, we conduct a sweep over $\alpha \in \{0.0, 0.2, 0.4, 0.6, 0.8, 1.0\}$ on Qwen-2.5-3B-Base. The results in Table 4 yield two key observations. (1) **Robust Improvement:** Stratified GRPO consistently outperforms the baseline GRPO ($\alpha = 0$) across all non-zero values. A high-performance plateau exists between $\alpha = 0.4$ and $\alpha = 0.8$, indicating that the method is not overly sensitive to the exact choice of $\alpha$. (2) **Benefit of Blending:** While the pure SAN variant ($\alpha = 1.0$) already achieves substantial gains over GRPO (36.7 vs. 33.3), the blended approach yields the best overall performance (41.6 at $\alpha = 0.6$). This empirically validates our theoretical motivation: blending preserves SAN's stratification while leveraging the global signal to stabilize finite-sample estimation.

## 6 CONCLUSION

In this work, we identify and formalize cross-stratum bias, a key obstacle for training LLM search agents with RL. It arises from improperly comparing structurally heterogeneous trajectories using a global baseline. It causes distorted credit assignment and hampers exploration. To address this, we introduce Stratified GRPO, a principled algorithm that partitions trajectories into homogeneous strata and computes advantages locally. Our analysis proves this method eliminates cross-stratum bias and achieves conditional unbiasedness and unit variance within each stratum, while retaining these properties globally. Extensive experiments on diverse QA and deep-research benchmarks demonstrate that Stratified GRPO substantially outperforms GRPO by up to 12.6 points, achieving higher rewards, greater training stability, and more effective search strategies. These results establish stratification as a principled remedy for structural heterogeneity in RL for LLM search agents.

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

---

**Algorithm 1:** Stratified GRPO

---

**Require:** Policy $p_\theta$, batch $B = \{\tau_1, \dots, \tau_K\}$ with rewards $\{R_i\}$, blending $\alpha \in [0, 1]$, stabilizer $\varepsilon > 0$.

1: Compute global stats: $\bar{R}_{\text{global}} \leftarrow \frac{1}{K} \sum_i R_i$ ;

   $\widehat{\sigma}_{\text{global}} \leftarrow \sqrt{\frac{1}{K} \sum_i (R_i - \bar{R}_{\text{global}})^2}$.

2: For all $i$, set $A_{\text{GN}}(\tau_i) \leftarrow (R_i - \bar{R}_{\text{global}})/(\widehat{\sigma}_{\text{global}} + \varepsilon)$.

3: Partition indices into per-prompt, per-stratum groups $I_k(x)$ (e.g., by search count).

4: **for** each prompt $x$ **do**

5:    **for** each stratum $k$ with index set $I_k(x)$ **do**

6:       $n_k \leftarrow |I_k(x)|$;    $\bar{R}_k \leftarrow \frac{1}{n_k} \sum_{i \in I_k(x)} R_i$;    $\widehat{\sigma}_k \leftarrow \sqrt{\frac{1}{n_k} \sum_{i \in I_k(x)} (R_i - \bar{R}_k)^2}$.

7:       **for** $i \in I_k(x)$ **do**

8:          $A_{\text{SAN}}(\tau_i) \leftarrow (R_i - \bar{R}_k)/(\widehat{\sigma}_k + \varepsilon)$.

9:          $A_{\text{blend}}(\tau_i) \leftarrow \alpha\, A_{\text{SAN}}(\tau_i) + (1 - \alpha)\, A_{\text{GN}}(\tau_i)$.

10:       **end for**

11:    **end for**

12: **end for**

13: Return gradient estimate $\widehat{g}_{\text{blend}} \leftarrow \frac{1}{K} \sum_{i=1}^{K} A_{\text{blend}}(\tau_i) \nabla_\theta \log p_\theta(\tau_i \mid x)$.

---

# A   PROOFS OF THEORETICAL RESULTS

## A.1   PROOF OF THEOREM 1

*Proof.* By construction, both advantage estimators are centered, meaning their sample mean over the batch $B$ is zero:

$$\frac{1}{K} \sum_{i=1}^{K} \hat{A}_G(\tau_i) = 0, \qquad \frac{1}{K} \sum_{i=1}^{K} \hat{A}_S(\tau_i) = \frac{1}{K} \sum_{k=0}^{I-1} \sum_{\tau_i \in B_k} (R_i - \bar{R}_k) = 0. \qquad (6)$$

The variance of the global advantage estimator is the total sample variance of the rewards:

$$\text{Var}[\hat{A}_G] = \frac{1}{K} \sum_{i=1}^{K} (R_i - \bar{R}_{\text{global}})^2.$$

We decompose this total variance by partitioning the sum by stratum and adding and subtracting the stratum mean $\bar{R}_k$ within the squared term:

$$\text{Var}[\hat{A}_G] = \frac{1}{K} \sum_{k=0}^{I-1} \sum_{\tau_i \in B_k} \left( (R_i - \bar{R}_k) + (\bar{R}_k - \bar{R}_{\text{global}}) \right)^2$$

$$= \frac{1}{K} \sum_{k=0}^{I-1} \sum_{\tau_i \in B_k} (R_i - \bar{R}_k)^2$$

$$+ \frac{1}{K} \sum_{k=0}^{I-1} \sum_{\tau_i \in B_k} 2(R_i - \bar{R}_k)(\bar{R}_k - \bar{R}_{\text{global}})$$

$$+ \frac{1}{K} \sum_{k=0}^{I-1} \sum_{\tau_i \in B_k} (\bar{R}_k - \bar{R}_{\text{global}})^2.$$

The cross-term in the above equation vanishes because, for each stratum $k$, the inner sum $\sum_{\tau_i \in B_k} (R_i - \bar{R}_k) = 0$ by the definition of $\bar{R}_k$. The above expression thus simplifies to the well-known variance decomposition formula:

$$\text{Var}[\hat{A}_G] = \underbrace{\frac{1}{K} \sum_{k=0}^{I-1} \sum_{\tau_i \in B_k} (R_i - \bar{R}_k)^2}_{\text{Within-Stratum Variance}} + \underbrace{\frac{1}{K} \sum_{k=0}^{I-1} n_k (\bar{R}_k - \bar{R}_{\text{global}})^2}_{\text{Between-Stratum Variance}}. \tag{7}$$

The first term in Equation (7) is precisely the definition of $\text{Var}[\hat{A}_S]$, since

$$\text{Var}[\hat{A}_S] = \frac{1}{K} \sum_{i=1}^{K} \left( \hat{A}_S(\tau_i) - \bar{\hat{A}}_S \right)^2,$$

where $\bar{\hat{A}}_S$ is the sample mean of the stratified advantages over the entire batch. As established in Equation (6), this mean is zero ($\bar{\hat{A}}_S = 0$). Therefore, the variance simplifies to the mean squared value:

$$\text{Var}[\hat{A}_S] = \frac{1}{K} \sum_{i=1}^{K} \left( \hat{A}_S(\tau_i) \right)^2.$$

To evaluate this sum, we partition it according to the strata $\{B_k\}_{k=0}^{I-1}$:

$$\text{Var}[\hat{A}_S] = \frac{1}{K} \sum_{k=0}^{I-1} \sum_{\tau_i \in B_k} \left( \hat{A}_S(\tau_i) \right)^2.$$

Finally, we substitute the definition of the stratified advantage, $\hat{A}_S(\tau_i) = R_i - \bar{R}_k$, for trajectories within each stratum $B_k$:

$$\text{Var}[\hat{A}_S] = \frac{1}{K} \sum_{k=0}^{I-1} \sum_{\tau_i \in B_k} (R_i - \bar{R}_k)^2.$$

This expression is identical to the first term in the variance decomposition identity (Equation (7)), thus proving the assertion.

The second term in Equation (7), the between-stratum variance, is a weighted sum of squares and is therefore non-negative. This yields the inequality:

$$\text{Var}[\hat{A}_G] = \text{Var}[\hat{A}_S] + \frac{1}{K} \sum_{k=0}^{I-1} n_k (\bar{R}_k - \bar{R}_{\text{global}})^2 \geq \text{Var}[\hat{A}_S]. \tag{8}$$

Equality holds if and only if the between-stratum variance component is zero. This requires every term in the sum to be zero, which means $\bar{R}_k = \bar{R}_{\text{global}}$ for all $k$. This condition is met only when all stratum means coincide. Otherwise, the between-stratum variance is strictly positive, and the inequality is strict. $\qquad \square$

**Remark 1.** *Equation* (7) *is the finite-sample form of the classical* within–between variance decomposition *in ANOVA. It shows that replacing the global baseline with stratum-specific baselines always weakly reduces the variance of advantage estimates, and strictly reduces it whenever the strata have heterogeneous reward means.*

### A.2 PROOF OF PROPOSITION 2

*Proof.* Let a stratum for a given prompt $x$ be denoted by $B_k(x)$, containing $n_k(x)$ trajectories with rewards $\{R_1, \ldots, R_{n_k(x)}\}$. The empirical mean and standard deviation are

$$\widehat{\mu}_k(x) = \frac{1}{n_k(x)} \sum_{i=1}^{n_k(x)} R_i$$

and

$$\widehat{\sigma}_k(x) = \sqrt{\frac{1}{n_k(x)} \sum_{i=1}^{n_k(x)} (R_i - \widehat{\mu}_k(x))^2}.$$

Without loss of generality, suppose that $n_k \geq 2$, and $\widehat{\sigma}_k(x) > 0$.

Consider the affine transformation $R_i' = aR_i + b$ for $a > 0$. We first compute the new empirical mean $\widehat{\mu}_k'(x)$ and standard deviation $\widehat{\sigma}_k'(x)$ for the transformed rewards.

The new mean is:

$$\widehat{\mu}_k'(x) = \frac{1}{n_k(x)} \sum_{i=1}^{n_k(x)} R_i' = \frac{1}{n_k(x)} \sum_{i=1}^{n_k(x)} (aR_i + b) = a\left(\frac{1}{n_k(x)} \sum_{i=1}^{n_k(x)} R_i\right) + b = a\widehat{\mu}_k(x) + b.$$

The new variance is:

$$(\widehat{\sigma}_k'(x))^2 = \frac{1}{n_k(x)} \sum_{i=1}^{n_k(x)} (R_i' - \widehat{\mu}_k'(x))^2$$

$$= \frac{1}{n_k(x)} \sum_{i=1}^{n_k(x)} \left((aR_i + b) - (a\widehat{\mu}_k(x) + b)\right)^2$$

$$= \frac{1}{n_k(x)} \sum_{i=1}^{n_k(x)} \left(a(R_i - \widehat{\mu}_k(x))\right)^2$$

$$= a^2 \left(\frac{1}{n_k(x)} \sum_{i=1}^{n_k(x)} (R_i - \widehat{\mu}_k(x))^2\right) = a^2(\widehat{\sigma}_k(x))^2.$$

Since $a > 0$, the new standard deviation is $\widehat{\sigma}_k'(x) = \sqrt{a^2(\widehat{\sigma}_k(x))^2} = a\widehat{\sigma}_k(x)$.

Finally, we compute the new advantage $A_{\text{SAN}}'$ for an arbitrary trajectory $\tau_i \in B_k(x)$:

$$A_{\text{SAN}}'(\tau_i) = \frac{R_i' - \widehat{\mu}_k'(x)}{\widehat{\sigma}_k'(x)}$$

$$= \frac{(aR_i + b) - (a\widehat{\mu}_k(x) + b)}{a\widehat{\sigma}_k(x)}$$

$$= \frac{a(R_i - \widehat{\mu}_k(x))}{a\widehat{\sigma}_k(x)}$$

$$= \frac{R_i - \widehat{\mu}_k(x)}{\widehat{\sigma}_k(x)} = A_{\text{SAN}}(\tau_i).$$

This shows that the advantage computed from the transformed rewards is identical to the original, completing the proof. □

### A.3 Proof of Theorem 2

*Proof.* We prove the identity by introducing the variance of the intermediate stratified estimator, $\text{Var}[\hat{A}_S]$, and decomposing the total difference into two parts. The total difference can be written as a telescoping sum:

$$\text{Var}[\hat{A}_G] - \text{Var}[A_{\text{SAN}}] = \big(\text{Var}[\hat{A}_G] - \text{Var}[\hat{A}_S]\big) + \big(\text{Var}[\hat{A}_S] - \text{Var}[A_{\text{SAN}}]\big). \qquad (9)$$

By Theorem 1,

$$\text{Var}[\hat{A}_G] - \text{Var}[\hat{A}_S] = \frac{1}{K} \sum_{k=0}^{I-1} n_k (\bar{R}_k - \bar{R}_{\text{global}})^2.$$

Moreover, by direct expansion, the empirical variance of the stratified advantage is

$$\text{Var}[\hat{A}_S] = \frac{1}{K} \sum_{k=0}^{I-1} n_k \widehat{\sigma}_k^2, \qquad \text{Var}[A_{\text{SAN}}] = \frac{1}{K} \sum_{k=0}^{I-1} n_k \frac{\widehat{\sigma}_k^2}{(\widehat{\sigma}_k + \varepsilon)^2}.$$

Then we can compute their difference:

$$\text{Var}[\hat{A}_S] - \text{Var}[A_{\text{SAN}}] = \frac{1}{K} \sum_{k=0}^{I-1} n_k \widehat{\sigma}_k^2 - \frac{1}{K} \sum_{k=0}^{I-1} n_k \frac{\widehat{\sigma}_k^2}{(\widehat{\sigma}_k + \varepsilon)^2} = \frac{1}{K} \sum_{k=0}^{I-1} n_k \widehat{\sigma}_k^2 \left(1 - \frac{1}{(\widehat{\sigma}_k + \varepsilon)^2}\right).$$

Substituting the two expressions into Equation (9) can get the final result. $\qquad \square$

### A.4 Proof of Theorem 3

Proving Theorem 3 relies on the following two lemmas:

**Lemma 1** (Conditional expectation of the score function). *Let $B \subseteq \{\tau\}$ be a fixed stratum and fix the context $x$. Assume $p_\theta(\tau \mid x)$ is differentiable in $\theta$, and that differentiation may be interchanged with summation/integration. Define*

$$Z(\theta; x) := \text{Pr}_\theta(\tau \in B \mid x) = \sum_{\tau \in B} p_\theta(\tau \mid x) \quad \text{(or the corresponding integral in the continuous case).}$$

$$(10)$$

*If $Z(\theta; x) > 0$, then*

$$\mathbb{E}[\nabla_\theta \log p_\theta(\tau \mid x) \mid \tau \in B, x] = \nabla_\theta \log \text{Pr}_\theta(\tau \in B \mid x).$$

*In particular, this conditional expectation equals zero if and only if $\text{Pr}_\theta(\tau \in B \mid x)$ is constant in $\theta$.*

*Proof.* By definition, the conditional distribution restricted to $B$ is

$$p_\theta(\tau \mid x, \tau \in B) = \frac{p_\theta(\tau \mid x)}{Z(\theta; x)}, \quad \tau \in B.$$

Hence, using $p \nabla \log p = \nabla p$ and assuming we can exchange differentiation and summation/integration,

$$\mathbb{E}[\nabla_\theta \log p_\theta(\tau \mid x) \mid \tau \in B, x] = \sum_{\tau \in B} p_\theta(\tau \mid x, \tau \in B) \nabla_\theta \log p_\theta(\tau \mid x)$$

$$= \frac{1}{Z(\theta; x)} \sum_{\tau \in B} p_\theta(\tau \mid x) \nabla_\theta \log p_\theta(\tau \mid x)$$

$$= \frac{1}{Z(\theta; x)} \sum_{\tau \in B} \nabla_\theta p_\theta(\tau \mid x)$$

$$= \frac{\nabla_\theta Z(\theta; x)}{Z(\theta; x)}$$

$$= \nabla_\theta \log Z(\theta; x),$$

where the sum can be replaced by an integral if $\tau$ is continuous. This proves the stated identity. The final claim about vanishing follows immediately. $\qquad \square$

**Remark 2.** *The identity above does* not *imply the conditional expectation is zero unless* $\Pr_\theta(\tau \in B \mid x)$ *is independent of* $\theta$. *If the stratum* $B$ *depends on* $\theta$ *(i.e.,* $B = B(\theta)$*), differentiating* $\Pr_\theta(\tau \in B(\theta) \mid x)$ *introduces additional terms due to the moving support; the above proof assumes* $B$ *is fixed.*

**Lemma 2** (Conditional Advantage–Score Identity). *For any stratum* $k$, *the following identity holds:*

$$\mathbb{E}_\theta\big[(R(\tau) - \mu_k(\theta))\, \nabla_\theta \log p_\theta(\tau \mid S = k) \,\big|\, S = k\big] = \nabla_\theta \mu_k(\theta), \tag{11}$$

*where* $\mu_k(\theta) := \mathbb{E}_\theta[R(\tau) \mid S = k]$ *is the mean reward in stratum* $k$.

*Proof.* This identity is a standard result from policy gradient theory, derived by applying the log-derivative trick to the gradient of an expectation and introducing the stratum mean $\mu_k(\theta)$ as a variance-reducing baseline. For detailed derivations, see the original REINFORCE paper by Williams (1992) and the textbook treatment in Sutton & Barto (2018). $\square$

Next, we give a proof of Theorem 3:

*Proof.* We prove the identity by first applying the law of total expectation to decompose the total expectation over the strata $k$:

$$\mathbb{E}_\theta\big[A_{\text{SAN}}(\tau)\, \nabla_\theta \log p_\theta(\tau)\big] = \sum_k \mathbb{E}_\theta\big[A_{\text{SAN}}(\tau)\, \nabla_\theta \log p_\theta(\tau)\, \mathbf{1}_{S=k}\big]$$

$$= \sum_k p_k(\theta)\, \mathbb{E}_\theta\Big[A_{\text{SAN}}(\tau)\, \nabla_\theta \log p_\theta(\tau) \,\big|\, S = k\Big]. \tag{12}$$

Now, we analyze the conditional expectation for a single stratum $k$. The key is to decompose the score function using the chain rule of probability, $p_\theta(\tau) = p_\theta(\tau \mid S = k)\, p_k(\theta)$. Taking the log-gradient gives the identity:

$$\nabla_\theta \log p_\theta(\tau) = \nabla_\theta \log p_\theta(\tau \mid S = k) + \nabla_\theta \log p_k(\theta).$$

Substituting this into the conditional expectation allows us to split it into two terms by linearity:

$$\mathbb{E}_\theta\Big[A_{\text{SAN}}(\tau)\, \nabla_\theta \log p_\theta(\tau) \,\big|\, S = k\Big]$$

$$= \underbrace{\mathbb{E}_\theta\Big[A_{\text{SAN}}(\tau)\, \nabla_\theta \log p_\theta(\tau \mid S = k) \,\big|\, S = k\Big]}_{\text{Term 1: Conditional Score Part}} + \underbrace{\mathbb{E}_\theta\Big[A_{\text{SAN}}(\tau)\, \nabla_\theta \log p_k(\theta) \,\big|\, S = k\Big]}_{\text{Term 2: Marginal Score Part}}.$$

We analyze each term separately.

**Term 2 (Marginal Score Part).** The term $\nabla_\theta \log p_k(\theta)$ depends only on the stratum index $k$ and the parameter $\theta$, so it is a constant with respect to the expectation conditional on $S = k$. We can therefore factor it out:

$$\mathbb{E}_\theta\Big[A_{\text{SAN}}(\tau)\, \nabla_\theta \log p_k(\theta) \,\big|\, S = k\Big] = (\nabla_\theta \log p_k(\theta)) \cdot \mathbb{E}_\theta[A_{\text{SAN}}(\tau) \mid S = k]$$

$$= (\nabla_\theta \log p_k(\theta)) \cdot \frac{\mathbb{E}_\theta[R(\tau) - \mu_k(\theta) \mid S = k]}{\sigma_k(\theta) + \varepsilon}$$

$$= (\nabla_\theta \log p_k(\theta)) \cdot \frac{\mu_k(\theta) - \mu_k(\theta)}{\sigma_k(\theta) + \varepsilon} = 0.$$

Thus, the second term vanishes exactly. This is the crucial step where the structural part of the gradient is eliminated.

**Term 1 (Conditional Score Part).** For this term, we substitute the definition of $A_{\text{SAN}}(\tau)$, noting that conditional on $S = k$, $\mu_S(\theta)$ becomes the constant $\mu_k(\theta)$:

$$\mathbb{E}_\theta\Big[A_{\text{SAN}}(\tau)\, \nabla_\theta \log p_\theta(\tau \mid S = k) \,\big|\, S = k\Big]$$

$$= \mathbb{E}_\theta\left[\frac{R(\tau) - \mu_k(\theta)}{\sigma_k(\theta) + \varepsilon}\, \nabla_\theta \log p_\theta(\tau \mid S = k) \,\bigg|\, S = k\right]$$

$$= \frac{1}{\sigma_k(\theta) + \varepsilon}\, \mathbb{E}_\theta\Big[(R(\tau) - \mu_k(\theta))\, \nabla_\theta \log p_\theta(\tau \mid S = k) \,\big|\, S = k\Big].$$

The remaining expectation is precisely the Conditional Advantage–Score Identity (Lemma 2), which equals $\nabla_\theta \mu_k(\theta)$. Therefore, the first term equals:

$$\frac{1}{\sigma_k(\theta) + \varepsilon} \nabla_\theta \mu_k(\theta).$$

**Conclusion.** Substituting the results for Term 1 and Term 2 back into the original sum over strata, we obtain the final result:

$$\mathbb{E}_\theta\big[A_{\mathrm{SAN}}(\tau)\,\nabla_\theta \log p_\theta(\tau)\big] = \sum_k p_k(\theta) \left( \frac{1}{\sigma_k(\theta) + \varepsilon} \nabla_\theta \mu_k(\theta) + 0 \right) = \sum_k \frac{p_k(\theta)}{\sigma_k(\theta) + \varepsilon} \nabla_\theta \mu_k(\theta).$$

This completes the proof. $\qquad\square$

## B  DETAILED STRUCTURAL COMPARISON OF SAN AND GLOBAL NORMALIZATION

In this section, we provide the rigorous algebraic proofs and structural analysis supporting the comparison summarized in Section 3.3. We offer a detailed structural comparison between Stratified Advantage Normalization (SAN) and the simpler, more common Global Normalization (GN). We demonstrate that GN, by ignoring trajectory heterogeneity, suffers from a structural flaw that SAN is designed to correct. Specifically, our analysis uses algebraic decomposition to pinpoint GN's cross-stratum offset bias and employs conditional statistics to prove that SAN serves as a pure, scale-stable learning signal.

**The Structural Flaw in Global Normalization.** The *global normalized* (GN) advantage defined in GRPO (Shao et al., 2024) et al. is

$$A_{\mathrm{GN}}(\tau_i) := \frac{R(\tau_i) - \bar{R}_{\mathrm{global}}}{\hat{\sigma}_{\mathrm{global}} + \varepsilon}. \tag{13}$$

Similar to Section 3.2, a core issue with GN is that it forces an "apples-to-oranges" comparison. This can be made precise by algebraically expressing the GN advantage in terms of the SAN advantage.

**Proposition 3** (Exact Advantage Decomposition). *For any fixed batch partition $\{B_k(x)\}_{k=0}^{I-1}$ and any $\tau_i \in B_k(x)$,*

$$A_{\mathrm{GN}}(\tau_i) = \underbrace{\frac{\hat{\sigma}_k(x) + \varepsilon}{\hat{\sigma}_{\mathrm{global}}(x) + \varepsilon}}_{:=\alpha_k(x)} A_{\mathrm{SAN}}(\tau_i) + \underbrace{\frac{\hat{\mu}_k(x) - \bar{R}_{\mathrm{global}}(x)}{\hat{\sigma}_{\mathrm{global}}(x) + \varepsilon}}_{:=\Delta_k(x)}. \tag{14}$$

*Proof.* The proof follows from direct algebraic manipulation. We start from the right-hand side (RHS) and substitute the definitions of $\alpha_k(x)$, $A_{\mathrm{SAN}}(\tau_i)$, and $\Delta_k(x)$:

$$\mathrm{RHS} = \alpha_k(x) \cdot A_{\mathrm{SAN}}(\tau_i) + \Delta_k(x)$$

$$= \left( \frac{\hat{\sigma}_k(x) + \varepsilon}{\hat{\sigma}_{\mathrm{global}}(x) + \varepsilon} \right) \cdot \left( \frac{R(\tau_i) - \hat{\mu}_k(x)}{\hat{\sigma}_k(x) + \varepsilon} \right) + \left( \frac{\hat{\mu}_k(x) - \bar{R}_{\mathrm{global}}(x)}{\hat{\sigma}_{\mathrm{global}}(x) + \varepsilon} \right) \quad \text{(by definition)}$$

$$= \frac{R(\tau_i) - \hat{\mu}_k(x)}{\hat{\sigma}_{\mathrm{global}}(x) + \varepsilon} + \frac{\hat{\mu}_k(x) - \bar{R}_{\mathrm{global}}(x)}{\hat{\sigma}_{\mathrm{global}}(x) + \varepsilon} \quad \text{(cancel } (\hat{\sigma}_k(x) + \varepsilon))$$

$$= \frac{(R(\tau_i) - \hat{\mu}_k(x)) + (\hat{\mu}_k(x) - \bar{R}_{\mathrm{global}}(x))}{\hat{\sigma}_{\mathrm{global}}(x) + \varepsilon} \quad \text{(combine terms)}$$

$$= \frac{R(\tau_i) - \bar{R}_{\mathrm{global}}(x)}{\hat{\sigma}_{\mathrm{global}}(x) + \varepsilon} \quad \text{(cancel } \hat{\mu}_k(x))$$

$$= A_{\mathrm{GN}}(\tau_i). \quad \text{(definition of } A_{\mathrm{GN}}) \quad \square$$

The decomposition in Proposition 3 reveals that the GN advantage equals a rescaled SAN advantage plus a **cross-stratum offset** ($\Delta_k$), which is the ultimate source of its systematic bias. This flaw carries over directly to the policy gradient.

**Gradient Bias from the Cross-Stratum Bias.** The decomposition in Equation (14) carries over directly to the gradient estimators. The GN gradient, $\widehat{g}_{\mathrm{GN}}$, decomposes into a SAN-like term and an additional bias-inducing term:

$$\widehat{g}_{\mathrm{GN}}(x) = \frac{1}{K} \sum_k \sum_{\tau_i \in B_k} \alpha_k \, A_{\mathrm{SAN}}(\tau_i) \, \nabla_\theta \log p_\theta(\tau_i \mid x) + \underbrace{\frac{1}{K} \sum_k \sum_{\tau_i \in B_k} \Delta_k \, \nabla_\theta \log p_\theta(\tau_i \mid x)}_{\text{Bias from Cross-Stratum Offset}}.$$

$$(15)$$

The decomposition in Equation (15) reveals a structural flaw in the GN gradient, driven by the cross-stratum offset $\Delta_k$. This term couples reward differences across strata with the policy's score vectors, introducing a systematic bias that persists whenever strata are heterogeneous. This fundamentally distorts the learning signal by forcing local credit assignment to depend on global statistics.

**Cross-Stratum Bias Hurts Exploration.** Because $\Delta_k$ in Equation (14) has the sign of $(\widehat{\mu}_k(x) - \bar{R}_{\mathrm{global}})$, the GN estimator systematically downweights strata whose mean reward lies below the global mean. As a result, GN may under-sample such strata even when they contain unexplored high-reward modes. This structural bias can therefore hinder exploration, consistent with our empirical findings in Figure 1 that GRPO cannot effectively explore policies that use more than one search call. Removing the offset is thus beneficial for promoting exploration.

**Analysis as a Signal Carrier.** To understand this more clearly, we analyze the conditional properties of the two estimators. An ideal "signal carrier" should be unbiased and have a consistent scale within each stratum, ensuring fair credit assignment. The following theorem formalizes why SAN provides a more reliable learning signal.

**Theorem 6** (Conditional Properties of SAN and GN Advantages. Theorem 4 in the main text). *Let $\varepsilon = 0$, and let the population reward statistics for a given prompt $x$ and stratum $k$ be defined as:*

$$\mu_k(x) := \mathbb{E}[R(\tau) \mid k, x], \qquad \sigma_k^2(x) := \mathrm{Var}(R(\tau) \mid k, x) \neq 0;$$
$$\mu(x) := \mathbb{E}[R(\tau) \mid x], \qquad \sigma^2(x) := \mathrm{Var}(R(\tau) \mid x) \neq 0.$$

*In the large-sample limit, the stratified (SAN) and global (GN) normalized advantages exhibit the following conditional properties for any stratum $k$:*

*1) **Conditional Expectation (Bias).** The SAN advantage is unbiased within each stratum, whereas the GN advantage carries a systematic bias proportional to the cross-stratum mean difference:*

$$\mathbb{E}[A_{\mathrm{SAN}} \mid k, x] = 0, \qquad \mathbb{E}[A_{\mathrm{GN}} \mid k, x] = \frac{\mu_k(x) - \mu(x)}{\sigma(x)}.$$

*2) **Conditional Variance.** The SAN advantage provides a consistent unit variance, whereas the GN advantage's variance scales with the ratio of stratum-to-global variance:*

$$\mathrm{Var}(A_{\mathrm{SAN}} \mid k, x) = 1, \qquad \mathrm{Var}(A_{\mathrm{GN}} \mid k, x) = \frac{\sigma_k^2(x)}{\sigma^2(x)}.$$

*Consequently, SAN acts as a* pure *and* scale-stable *signal carrier, while GN introduces a cross-stratum bias and inconsistent scaling whenever reward heterogeneity is present.*

*Proof.* The proof proceeds by direct calculation of the conditional expectation and variance for each estimator in the large-sample limit. All quantities are conditioned on a fixed prompt $x$.

First, we prove the results for $A_{\mathrm{SAN}}$. The SAN advantage for a trajectory $\tau$ in stratum $k$ is defined as $A_{\mathrm{SAN}}(\tau) = \frac{R(\tau) - \mu_k}{\sigma_k}$ (ignoring $\varepsilon$ for clarity in the limit). By the linearity of expectation,

$$\mathbb{E}[A_{\mathrm{SAN}}(\tau) \mid k, x] = \mathbb{E}\left[ \frac{R(\tau) - \mu_k}{\sigma_k} \,\middle|\, k, x \right]$$
$$= \frac{1}{\sigma_k} \big( \mathbb{E}[R(\tau) \mid k, x] - \mu_k \big)$$
$$= \frac{1}{\sigma_k} (\mu_k - \mu_k) = 0.$$

This shows that SAN is an unbiased signal carrier within each stratum. Next, by the properties of variance, $\mathrm{Var}(cX + d) = c^2 \, \mathrm{Var}(X)$,

$$\mathrm{Var}(A_{\mathrm{SAN}}(\tau) \mid k, x) = \mathrm{Var}\left(\frac{R(\tau) - \mu_k}{\sigma_k} \,\Big|\, k, x\right)$$

$$= \frac{1}{\sigma_k^2} \, \mathrm{Var}(R(\tau) - \mu_k \mid k, x)$$

$$= \frac{1}{\sigma_k^2} \, \mathrm{Var}(R(\tau) \mid k, x)$$

$$= \frac{\sigma_k^2}{\sigma_k^2} = 1.$$

This shows that SAN provides a consistently scaled (unit variance) carrier in every stratum.

Second, we prove the results for $A_{\mathrm{GN}}$. The GN advantage is defined as $A_{\mathrm{GN}}(\tau) = \frac{R(\tau) - \mu}{\sigma}$. So

$$\mathbb{E}[A_{\mathrm{GN}}(\tau) \mid k, x] = \mathbb{E}\left[\frac{R(\tau) - \mu}{\sigma} \,\Big|\, k, x\right]$$

$$= \frac{1}{\sigma}\big(\mathbb{E}[R(\tau) \mid k, x] - \mu\big)$$

$$= \frac{\mu_k - \mu}{\sigma}.$$

This is generally non-zero whenever stratum means differ ($\mu_k \neq \mu$), which confirms that GN is a biased carrier within the stratum. This non-zero expectation represents a spurious signal. Next,

$$\mathrm{Var}(A_{\mathrm{GN}}(\tau) \mid k, x) = \mathrm{Var}\left(\frac{R(\tau) - \mu}{\sigma} \,\Big|\, k, x\right)$$

$$= \frac{1}{\sigma^2} \, \mathrm{Var}(R(\tau) - \mu \mid k, x)$$

$$= \frac{1}{\sigma^2} \, \mathrm{Var}(R(\tau) \mid k, x) = \frac{\sigma_k^2}{\sigma^2}.$$

This shows that the variance of the GN carrier is not consistent across strata; it depends on the ratio of the stratum's variance to the global variance.

The analysis of the conditional statistics reveals the structural superiority of SAN. Within any given stratum, SAN provides a zero-mean, unit-variance signal carrier, which serves as a pure and consistent baseline for the policy gradient calculation. GN, in contrast, introduces a spurious signal (a non-zero mean) and has an inconsistent variance, making it a biased and less reliable signal carrier. This completes the proof. $\square$

Theorem 6 shows SAN's structural advantage: it enforces zero-mean, unit-variance signals within each stratum, ensuring fair credit assignment relative to peers. GN, by contrast, introduces a spurious offset (nonzero conditional mean) and distorts scaling across strata.

Having established their conditional properties, we now analyze the global (marginal) moments of the two estimators.

**Theorem 7** (Global Moments of SAN and GN. Theorem 5 in the main text). *Let $\varepsilon = 0$. Fix a prompt $x$ and let $S \in \{0, \ldots, I - 1\}$ denote the stratum index with mixing weights $p_k(x) = \mathbb{P}(S = k \mid x)$. Write the population reward moments as in Theorem 6:*

$$\mu_k(x) = \mathbb{E}[R \mid S{=}k, x], \quad \sigma_k^2(x) = \mathrm{Var}(R \mid S{=}k, x) \neq 0;$$

$$\mu(x) = \mathbb{E}[R \mid x], \quad \sigma^2(x) = \mathrm{Var}(R \mid x) \neq 0.$$

*Consider the large-sample (population)* SAN *and* GN *advantages:*

$$A_{\mathrm{SAN}} = \frac{R - \mu_S(x)}{\sigma_S(x)}, \qquad A_{\mathrm{GN}} = \frac{R - \mu(x)}{\sigma(x)}.$$

*Then, (a) **Global Means:** $\mathbb{E}[A_{\mathrm{SAN}} \mid x] = 0$, $\mathbb{E}[A_{\mathrm{GN}} \mid x] = 0$.*

*(b) **Global Variances:** $\mathrm{Var}(A_{\mathrm{SAN}} \mid x) = \mathrm{Var}(A_{\mathrm{GN}} \mid x) = 1$.*

*Proof.* **For (a)**, applying the law of total expectation,

$$\mathbb{E}[A_{\mathrm{SAN}} \mid x] = \sum_k p_k(x) \, \mathbb{E}\left[\frac{R - \mu_k(x)}{\sigma_k(x)} \,\middle|\, S=k, x\right]$$

$$= \sum_k p_k(x) \, \frac{\mathbb{E}[R - \mu_k(x) \mid S=k, x]}{\sigma_k(x)}$$

$$= 0.$$

$$\mathbb{E}[A_{\mathrm{GN}} \mid x] = \frac{\mathbb{E}[R \mid x] - \mu(x)}{\sigma(x)} = 0.$$

**For (b)**, by the law of total variance,

$$\mathrm{Var}(A_{\mathrm{SAN}} \mid x) = \mathbb{E}[\mathrm{Var}(A_{\mathrm{SAN}} \mid S, x)] + \mathrm{Var}(\mathbb{E}[A_{\mathrm{SAN}} \mid S, x]).$$

We now analyze each of the two terms in the right-hand side separately.

From Theorem 6, we know that the idealized SAN advantage is conditionally unbiased in every stratum:

$$\mathbb{E}[A_{\mathrm{SAN}} \mid S = k, x] = 0, \quad \text{for all strata } k.$$

Since the conditional expectation is a constant (zero) regardless of the stratum $S$, its variance is zero:

$$\mathrm{Var}(\mathbb{E}[A_{\mathrm{SAN}} \mid S, x]) = \mathrm{Var}(0) = 0.$$

Next, we evaluate the first term, which is the expected variance within strata. We start with the inner term, the variance conditional on a specific stratum $k$:

$$\mathrm{Var}(A_{\mathrm{SAN}} \mid S = k, x) = \mathrm{Var}\left(\frac{R - \mu_k(x)}{\sigma_k(x)} \,\middle|\, k, x\right)$$

$$= \frac{1}{\sigma_k^2(x)} \, \mathrm{Var}(R \mid k, x)$$

$$= \frac{\sigma_k^2(x)}{\sigma_k^2(x)} = 1.$$

Now, we take the expectation of this quantity over the random stratum $S$. This is equivalent to a weighted average over all strata, where the weights are the probabilities $p_k(x) := P(S = k \mid x)$:

$$\mathbb{E}[\mathrm{Var}(A_{\mathrm{SAN}} \mid S, x)] = \sum_k p_k(x) \cdot \mathrm{Var}(A_{\mathrm{SAN}} \mid S = k, x)$$

$$= \sum_k p_k(x) \frac{\sigma_k^2(x)}{\sigma_k^2(x)} = \sum_k p_k(x) = 1.$$

Finally, we add the two components back together:

$$\mathrm{Var}(A_{\mathrm{SAN}} \mid x) = \mathbb{E}[\mathrm{Var}(A_{\mathrm{SAN}} \mid S, x)] + \mathrm{Var}(\mathbb{E}[A_{\mathrm{SAN}} \mid S, x]) = 1 + 0 = 1.$$

For GN, since $A_{\mathrm{GN}} = (R - \mu(x))/\sigma(x)$,

$$\mathrm{Var}(A_{\mathrm{GN}} \mid x) = \frac{\mathrm{Var}(R \mid x)}{\sigma^2(x)} = \frac{\sigma^2(x)}{\sigma^2(x)} = 1. \qquad \square$$

Theorem 7 shows that despite their different mechanisms: SAN normalizing locally within strata and GN normalizing globally, both idealized estimators are **globally unbiased** (mean zero) and achieve the **exact same unit variance** in the idealized case where $\varepsilon = 0$.

The comparison of moments in Theorems 6 and 7 is summarized in Appendix C. Theorems 6 and 7 reveal a crucial distinction between the global (overall) and conditional (within-stratum) properties of the idealized SAN and GN estimators. Globally, the two estimators appear equivalent: Theorem 7 shows that both are unbiased (mean zero) and, in the ideal case where $\varepsilon = 0$, have the exact same unit variance. However, this global equivalence masks a fundamental structural difference at the conditional level, which is what truly governs the learning dynamics. Theorem 6 establishes that SAN is conditionally pure: it delivers a zero-mean, unit-variance signal within every single stratum. In stark contrast, GN is conditionally biased and has inconsistent scaling, meaning its mean and variance fluctuate from one stratum to another.

## C  COMPARISON OF LOCAL VS. GLOBAL MOMENTS OF SAN AND GN ADVANTAGES

Table 5: Local (conditional on stratum $S{=}k$) vs. global (marginal over $S$) moments of SAN and GN advantages. Here $\sigma_k^2 = \mathrm{Var}(R \mid S{=}k, x)$ and $\sigma^2 = \mathrm{Var}(R \mid x)$. For $\varepsilon = 0$, SAN achieves joint standardization (local & global), whereas GN only standardizes globally.

|  | Local (given $S{=}k$) | Global (marginal over $S$) |
|---|:---:|:---:|
| SAN mean | 0 | 0 |
| SAN variance | 1 | 1 |
| GN mean | $\frac{\mu_k - \mu}{\sigma_2}$ | 0 |
| GN variance | $\frac{\sigma_k^2}{\sigma^2}$ | 1 |

## D  FURTHER ANALYSIS OF STRATIFIED GRPO

**Multi-seed Stability Analysis.**  We evaluate the robustness and stability of Stratified GRPO by conducting the factual QA experiment on Qwen 2.5 3B Base using three random seeds. As shown in Table 6, Stratified GRPO yields consistent gains across all seeds, underscoring the method's reliability. We visualize the training dynamics averaged across these independent runs in Figure 2 and Figure 3. Figure 2 demonstrates that Stratified GRPO achieves consistently higher training rewards and effective multi-turn search policies, alongside lower and smoother gradient norms and KL divergence compared to GRPO. In Figure 3, we observe that Stratified GRPO successfully expands the average rollout length, whereas GRPO collapses to short responses, aligning with the "Search Calls" plot and confirming that our method encourages long-horizon exploration. While entropy converges at a similar rate for both methods, Stratified GRPO converges to a higher-reward policy. Regarding advantage estimates, the global normalized advantage shows unit variance for both methods, supporting Theorem 5. Crucially, the in-stratum normalized advantage variance confirms that Stratified GRPO maintains unit variance within each stratum while GRPO does not, empirically validating Theorem 4. Furthermore, Stratified GRPO's in-stratum normalized advantage variance is consistently lower than GRPO, further demonstrating our method's stability. Finally, Stratified GRPO exhibits lower global unnormalized advantage variance than GRPO, empirically validating Theorem 1 and further attesting to the stability of our approach.

Table 6: Stability analysis by training Qwen2.5-3B-Base using 3 random seeds for GRPO and our Stratified GRPO.

| Method | NQ | TriviaQA | PopQA | HotpotQA | 2Wiki | Musique | Bamboogle | Avg. |
|---|---|---|---|---|---|---|---|---|
| GRPO | $44.5 \pm 1.0$ | $60.9 \pm 0.4$ | $43.4 \pm 0.4$ | $32.1 \pm 0.5$ | $29.2 \pm 0.4$ | $8.0 \pm 0.2$ | $14.0 \pm 0.9$ | $33.2 \pm 0.2$ |
| Stratified GRPO | $45.8 \pm 0.2$ | $61.6 \pm 0.6$ | $44.0 \pm 0.9$ | $41.1 \pm 0.4$ | $39.3 \pm 1.4$ | $17.4 \pm 0.7$ | $38.7 \pm 3.5$ | $41.1 \pm 0.5$ |

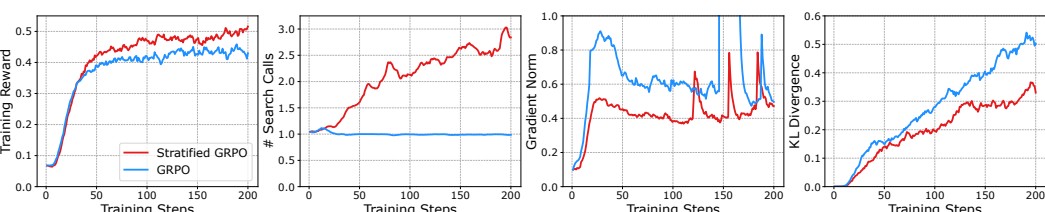

Figure 2: Training dynamics comparing Stratified GRPO to GRPO on Qwen 2.5 3B Base, with results averaged across three individual runs. The four plots show (1) training reward, (2) search calls per question, (3) gradient norms, and (4) KL divergence over training.

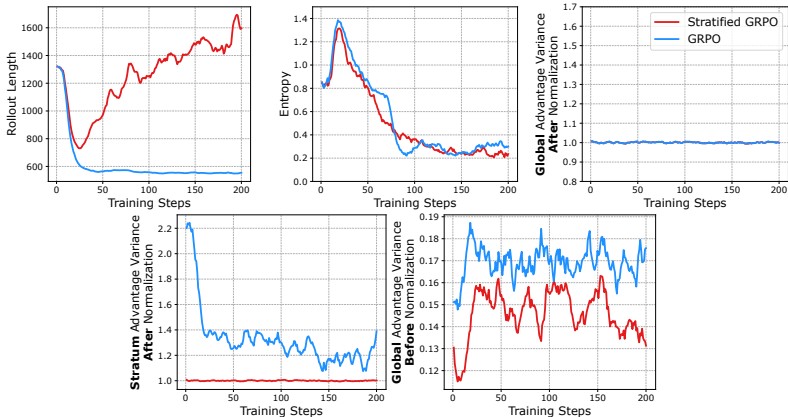

Figure 3: Training dynamics comparing Stratified GRPO to GRPO on Qwen 2.5 3B Base, with results averaged across three individual runs. The five plots show (1) rollout length, (2) entropy, (3) global normalized advantage variance, (4) in-stratum normalized advantage variance, and (5) global unnormalized advantage variance.

Table 7: Validation of the stratification heuristic by comparing our stratification heuristic based on tool calling numbers with random stratification.

| Method | NQ | TriviaQA | PopQA | HotpotQA | 2Wiki | Musique | Bamboogle | Avg. |
|---|---|---|---|---|---|---|---|---|
| GRPO | 45.2 | 61.2 | 43.8 | 32.6 | 29.7 | 7.8 | 12.9 | 33.3 |
| randomly assign 2 strata | 43.9 | 60.7 | 43.9 | 31.6 | 29.4 | 7.8 | 14.5 | 33.1 |
| randomly assign 3 strata | 44.1 | 61.4 | 43.6 | 32.5 | 28.0 | 7.7 | 15.3 | 33.2 |
| Stratified GRPO | 45.9 | 61.4 | 43.0 | 40.8 | 39.9 | 17.7 | 42.7 | 41.6 |

**Validation of the Stratification Heuristic.** In Section 3.2, we identified the number of tool calls as the primary driver of heterogeneity in LLM tool-use trajectories, motivating its use as our stratification heuristic. To validate this design choice, we introduce a control experiment where sampled rollouts are randomly assigned to arbitrary strata ($K = 2$ or $3$), while keeping all other hyperparameters the same as our method. As shown in Table 7, random stratification yields performance nearly identical to the baseline GRPO (33.1–33.2 vs. 33.3 average). In contrast, our proposed Stratified GRPO improves the average by +8.3 points, with particularly significant gains on multi-hop datasets. This result confirms that the performance benefits stem from structure-aware stratification rather than simple trajectory partitioning. Furthermore, this empirical finding aligns with our theoretical framework: when strata are assigned randomly, stratum means converge to the global mean (up to sampling noise). Consequently, as implied by Theorem 1, random assignment fails to reduce variance or mitigate bias.

# E  EXPERIMENT DETAILS

## E.1  TRAINING SETTINGS

For factual QA tasks, we mainly follow the settings of Search-R1 (Jin et al., 2025). We train our models on 8 GPUs using a global batch size of 256 and a mini-batch size of 256. The maximum sequence length is set to 4096 tokens, with maximum response length and retrieved content length of 500 tokens in each interaction turn. For rollout sampling, we use a temperature of 1.0 and a top-p value of 1.0. We use a learning rate of 1e-6 with a warm-up ratio of 0.1. Training is conducted for 200 steps. We use a KL divergence coefficient $\beta$ of 0.001 and a clipping ratio $\epsilon$ of 0.2. For both GRPO and Stratified GRPO, we sample 8 responses per prompt. Stratified GRPO uses $\alpha$ of 0.8 for Qwen 2.5 3B Instruct and 0.6 for Qwen 2.5 3B Base. The number of maximum interaction turns is set to 4, and we retrieve the top 3 passages for each search call. Our implementation is based on the Verl framework (Sheng et al., 2025) For deep-research agent tasks, we follow the settings of VerlTool (Jiang et al., 2025) and base our implementation on the VerlTool framework.

## F  REPRODUCIBILITY STATEMENT

To support the reproducibility of Stratified GRPO, we have discussed implementation details and key resources in the main text and appendix in detail. All algorithmic details are described in Section 3 and Algorithm 1. Full training and evaluation details are specified in Section 5 and Section E. All benchmarks (NQ, TriviaQA, PopQA, HotpotQA, 2WikiMultiHopQA, MuSiQue, Bamboogle, and GAIA) and base models (Qwen-2.5-3B Base/Instruct and Qwen3-8B) are publicly available. We report ablations, sensitivity analyses, and multi-seed stability experiments in Section 5.3 and Section D. Upon acceptance, we will release our full codebase, configuration files, scripts, and trained model checkpoints to facilitate validation of our results.

## G  LLM USAGE DISCLOSURE

We utilized a large language model (LLM) as a writing assistant to enhance the clarity, grammar, and readability of this manuscript. The LLM's role was limited to tasks such as rewording sentences, refining technical descriptions, and formatting tables into LaTeX. All scientific content—including the conceptual framework, analysis, experiments, and conclusions—was developed exclusively by the human authors. The authors have meticulously reviewed all LLM-generated suggestions and bear full responsibility for the scientific integrity and final content of this work.

