# OpenReview forum: "Stratified GRPO: Handling Structural Heterogeneity in Reinforcement Learning of LLM Search Agents"
_ICLR.cc/2026/Conference — Submitted to ICLR 2026_

### Official Review · Reviewer_ukKu · 2025-10-15

**Soundness:** 3
**Presentation:** 2
**Contribution:** 3
**Rating:** 4
**Confidence:** 5

**Summary:**

This paper addresses a critical yet often overlooked challenge in training large language model (LLM) search agents with reinforcement learning: structural heterogeneity in trajectories—where differences in the number, placement, or outcomes of search calls lead to fundamentally distinct behavior patterns and reward distributions. The authors show that standard policy gradient methods like GRPO, which rely on a single global baseline to compute advantages, suffer from cross-stratum bias: an “apples-to-oranges” comparison across structurally incomparable trajectories that distorts credit assignment and hinders exploration of complex, multi-hop search strategies. To resolve this, they propose Stratified GRPO, centered on Stratified Advantage Normalization (SAN), which partitions trajectories into homogeneous strata based on structural features (e.g., search count) and computes normalized advantages within each stratum. Theoretical analysis proves that SAN eliminates cross-stratum bias, yielding conditionally unbiased, unit-variance advantages per stratum while preserving global unbiasedness and unit variance. For improved stability in finite-sample settings, they further introduce Blended Advantage, a linear combination of SAN and global normalization (GN). Experiments across seven question-answering benchmarks—spanning both single-hop and multi-hop tasks—show that Stratified GRPO outperforms GRPO by up to 11.3 EM points on average, with especially strong gains on multi-hop problems, along with higher training rewards, greater stability, and more effective search policies that actively issue multiple queries.

**Strengths:**

- **Precise and practically relevant problem identification**: The paper is the first to formally identify and define *cross-stratum bias*—a pervasive structural flaw in LLM tool-use scenarios—thereby filling a critical theoretical gap in reinforcement learning for LLM agents.
- **Simple yet principled method**: Stratified Advantage Normalization (SAN) is grounded in classical stratification from statistics. It requires no changes to the policy architecture or additional models; the core issue is resolved solely through a refined normalization of the advantage function.

**Weaknesses:**

- **overemphasizes “theoretical”**:   I argue that the paper overemphasizes “theoretical” while the experimental evidence lacks sufficient persuasiveness. For instance, Theorem 1 claims that Stratified GRPO reduces variance—but do the experiments actually support this theoretical conclusion? The authors should provide direct empirical evidence of more stable training, such as smoother gradient norm (grad-norm) curves, more stable KL divergence trajectories, or comparative plots showing reduced within-stratum variance of advantage estimates over training.
Moreover, Proposition 2 (Invariance to Positive Affine Reward Transforms) is a property shared by all GRPO-based methods that use reward normalization—it is not unique to Stratified GRPO. Highlighting it as a distinctive advantage is therefore misleading and inflates the perceived novelty of the method.

- **Unconvincing experiments**:  The reward curves shown in Figure 1 are highly counterintuitive—the Instruct model performs significantly worse than the Base model during training. This strongly suggests a potential engineering issue, such as suboptimal hyperparameter settings or implementation details. Given that the authors have not released their code and the reported experimental details are insufficient, I believe the current reward curves alone cannot convincingly demonstrate a methodological advantage of Stratified GRPO. Instead, they appear more like the outcome of an inadequately tuned baseline (e.g., GRPO on the Instruct model).
Reinforcement learning training is inherently high-variance and prone to instability. To substantiate their claims, the authors should report the variance across multiple random seeds and provide training curves averaged over several independent runs—not just a single trajectory. Without such evidence, it is difficult to distinguish genuine algorithmic improvement from favorable random fluctuations or inconsistent tuning.

- **Stratification relies on handcrafted heuristics**: The current approach partitions trajectories by "number of search calls," which is heuristic and task-specific. The paper does not explore more general or automated stratification mechanisms (e.g., clustering based on behavioral embeddings). If the true source of structural heterogeneity lies in query semantics or retrieval quality, this simple partitioning may be insufficient.

- **Experiments are limited to question-answering tasks**: All evaluations are conducted on Wikipedia-based retrieval QA benchmarks. The method’s generalizability to other tool-augmented settings—such as code generation, API calling, or multi-tool coordination—remains unverified.

- **Lack of comparison with structure-aware critic baselines**: For instance, a PPO variant using a conditional value function like \(V(s, \text{search\_count})\) might similarly mitigate cross-stratum bias. Without such comparisons, it is difficult to claim that SAN is the only—or optimal—solution to the problem.

**Questions:**

The authors are encouraged to provide results from multiple independent runs, along with hyperparameter sensitivity analyses. Additionally, more training curves and diagnostic metrics should be included to strengthen the empirical support for the proposed method. These curves may include, but are not limited to:
- Policy entropy over time,
- Gradient norm dynamics,
- Variance of importance weights,
- Distribution or average of rollout lengths,
- Variance of advantage estimates (both overall and within each stratum),
- KL divergence between the current and reference policies.

For all reported metrics, both the mean and variance across random seeds should be presented. Crucially, any experimental evidence that directly validates the theoretical claims—such as reduced within-stratum advantage variance or more stable credit assignment—should be explicitly highlighted. Providing this comprehensive set of analyses would establish a tight **theory–experiment feedback loop**, significantly enhancing the credibility and persuasiveness of the theoretical contributions.

---

> ### Author Response · Authors · 2025-11-22
> **Response 1**
>
> **Q1 “Overemphasizes theory” and requests empirical evidence for stability**
>
> We note that the original Figure 1 already exhibits the relevant qualitative pattern: Stratified GRPO achieves higher rewards and avoids the instability and collapse observed with GRPO. In response to the reviewer’s request, we now confirm and quantify this effect by reporting training dynamics averaged over three independent runs (new Figure 2 in the appendix). The figure includes training reward, number of search calls, gradient norms, and KL divergence trajectories.
>
> 1.  **Training reward.** Stratified GRPO attains consistently higher rewards throughout training and yields significantly better evaluation performance than GRPO. The evaluation performance is shown in our response to **Q4**, averaged over three seeds.
>
> 2.  **Search calls per question.** Stratified GRPO steadily increases the number of search calls over time, whereas GRPO stays near one call per question. This indicates that SAN specifically encourages exploration of effective multi-turn search behaviors, rather than collapsing to a single-turn strategy.
>
> 3.  **Gradient norms.** GRPO produces larger and more erratic gradient norms, with pronounced spikes. In contrast, Stratified GRPO maintains lower and more controlled gradients, reflecting substantially more stable updates.
>
> 4.  **KL divergence.** Stratified GRPO exhibits consistently lower KL divergence from the reference policy than GRPO, indicating a more stable and less brittle optimization trajectory.
>
>
> Finally, SAN guarantees zero-mean, unit-variance advantages within each stratum by construction (Theorem 4), and in the idealized limit both SAN and GN are globally unit-variance (Theorem 5). Therefore, gradient-norm and KL trajectories are the appropriate empirical diagnostics for stability and variance reduction. The new multi-seed dynamics in Figure 2 provide direct evidence supporting the theoretical mechanism.
>
>
> **Q2 Proposition 2 is not unique to SAN**
>
> We acknowledge that this property is shared by all normalized advantages. We included Proposition 2 not as a claim of uniqueness, but to emphasize that SAN successfully implements the desirable **scale-stability** property of normalization _within_ the stratified framework, which is a necessary component for the overall utility of SAN and the Blended Advantage. The true structural advantage of SAN remains its **conditional purity** (zero mean and unit variance per stratum) as proven in **Theorem 4**, which GN fundamentally lacks.
>
> **Q3 Counterintuitive experiments: Figure 1 shows the Instruct model training worse than the Base model**
>
> We respectfully disagree that this indicates an experimental flaw. Our setup follows established practice in LLM search-agent RL, specifically Search-R1 [1]. We implement our method on their official codebase and report full experiment details in Section 4.1 and Appendix D. The phenomenon that the Base model trains better than the Instruct model is also observed in Search-R1. Their Tables 2, 3, and 5 demonstrate that the Base models usually perform better than Instruct models. Their released wandb training curves in their official github repo show similar trends, including collapse behavior for the Instruct model.
>
> Thus, this is not suggestive of an engineering issue or an under-tuned baseline. Importantly, Stratified GRPO improves **both** Base and Instruct models, and in particular prevents GRPO’s collapse on the Instruct model. This supports an algorithmic explanation rather than a tuning artifact.
>
> We will release the full code upon acceptance.
>
> [1] _Search-R1: Training LLMs to Reason and Leverage Search Engines with Reinforcement Learning._ COLM 2025.

---

> ### Author Response · Authors · 2025-11-22
> **Response 2**
>
> **Q4 Multi-seed results**
>
> We agree that reporting variability across random seeds is valuable. At the same time, we want to clarify the experiment norm in large-scale LLM agent RL: due to the extreme computational cost of post-training with on-policy rollouts, the standard practice of contemporary LLM search agent RL papers is to report a single training run [1]. This is not unique to our work, but rather common practice in this field.
>
> Nevertheless, to address the concern directly, we have added multi-seed experiments in the following table. Specifically, we train Qwen2.5-3B-Base using 3 seeds for GRPO and our Stratified GRPO and report the mean $\pm$ std over independent runs for the seven benchmarks.
>
> The gains of Stratified GRPO are consistent across seeds, strengthening the robustness of our method. In addition, the averaged training curves in new Figure 2 of the appendix further show that Stratified GRPO reliably yields higher training reward  and consistently incentivizes effective multi-turn search policies.
>
> | Method | nq | triviaqa | popqa | hotpotqa | 2wikimultihopqa | musique | bamboogle | avg |
> |---|---:|---:|---:|---:|---:|---:|---:|---:|
> | GRPO | 44.5 $\pm$ 1.0 | 60.9 $\pm$ 0.4 | 43.4 $\pm$ 0.4 | 32.1 $\pm$ 0.5 | 29.2 $\pm$ 0.4 | 8.0 $\pm$ 0.2 | 14.0 $\pm$ 0.9 | 33.2 $\pm$ 0.2 |
> | Ours | 45.8 $\pm$ 0.2 | 61.6 $\pm$ 0.6 | 44.0 $\pm$ 0.9 | 41.1 $\pm$ 0.4 | 39.3 $\pm$ 1.4 | 17.4 $\pm$ 0.7 | 38.7 $\pm$ 3.5 | 41.1 $\pm$ 0.5 |
>
>
>
>
>
> **Q5 Sensitivity analysis of the hyperparameter $\alpha$.**
>
> We appreciate your suggestion of sensitivity analysis of the hyperparameter $\alpha$. Below we report a detailed sweep over $\alpha \in \\{0.0, 0.2, 0.4, 0.6, 0.8, 1.0\\}$ on Qwen-2.5-3B-Base:
>
> |  value of $\alpha$  | NQ | TriviaQA | PopQA | HotpotQA | 2Wiki | Musique | Bamboogle |Avg  |
> |---|---:|---:|---:|---:|---:|---:|---:|---:|
> | 0.0 (GRPO) | 45.2 | 61.2 | 43.8 | 32.6 | 29.7 | 7.8| 12.9 | 33.3 |
> | 0.2 | 45.1 | 61.8 | 43.1 | 32.2 | 29.5 | 7.6 | 14.5 | 33.4 |
> | 0.4 | 44.6 | 59.8 | 43.2 | 39.6 | 36.5 | 15.6 | 39.5 | 39.8 |
> | 0.6 | 45.9 | 61.4 | 43.0 | 40.8 | 39.9 | 17.7 | 42.7 | 41.6 |
> | 0.8 | 45.1 | 60.1 | 43.9 | 40.3 | 41.9 | 17.5 | 37.9 | 41.0 |
> | 1.0 (SAN-only) | 43.7 | 59.3 | 41.1 | 36.6 | 38.4 | 12.6 | 25.0 | 36.7 |
>
>
> Key observations:
>
> -   **Robust improvement over GRPO.** All $\alpha \ge 0.4$ significantly outperform GRPO, with a broad and stable high-performance region between 0.4 and 0.8. This indicates that our method is not sensitive to the exact value of $\alpha$.
>
> -   **SAN-only is already strong but blending is better. The** $\alpha = 1.0$ (SAN-only) variant already yields substantial gains over GRPO on average, especially on multi-hop datasets. However, advantage blending yields the best overall trade-off, as our theory suggests.
>
> Overall, these results confirm the robustness and stability of our Stratified GRPO, which achieves stable high-performance with broad  region of $\alpha$ values.
>
>
>
>
> **Q6 Stratification relies on handcrafted heuristics (search count)**
>
> We agree that stratification quality matters, but the concern here seems to conflate our **instantiation** with the **method**. Stratified GRPO / SAN is _not_ tied to search count nor to any task-specific heuristic. Definition 1 and Algorithm 1 explicitly permit **any discrete partitioning function over structural features**; search count is presented only as a canonical example for search agents. The theoretical guarantees (bias removal and variance decomposition) require only that trajectories within a stratum are structurally comparable — they do **not** assume a particular choice of stratum.
>
> In the setting of search agents, the number of tool calls is not an arbitrary heuristic, it is the most direct observable driver of structural heterogeneity. Trajectories with different numbers of search calls are qualitatively different, induce different behavior modes, and follow distinct reward distributions, which is precisely the setting SAN is designed for. This choice therefore isolates the phenomenon of cross-stratum bias in the clearest possible way, without introducing additional modeling assumptions.
>
> The suggestions (e.g., clustering by semantic embeddings or retrieval quality) are interesting **orthogonal extensions**. They effectively propose learning a stratification function, which is a separate research problem and not required to test whether stratified advantage estimation eliminates cross-stratum bias once heterogeneity is present. Exploring automatic strata would also introduce new design degrees of freedom (representation, distance metrics, clustering stability) that could confound the paper’s main contribution. We view these as valuable future directions.
>
> [1] _Search-R1: Training LLMs to Reason and Leverage Search Engines with Reinforcement Learning._ COLM 2025.

---

> ### Author Response · Authors · 2025-11-22
> **Response 3**
>
> **Q7 Experiments are limited to question-answering tasks.**
>
> We agree that broader validation is valuable. Our current experiments focus on factual QA, following Search-R1 [1]. However, **our algorithmic design is domain-agnostic**: the theory is derived for a general discrete stratum variable $S$ and does not assume QA-specific structure.
>
> To further evaluate generality, we add experiments on **tool-use agents in real-world multi-step settings**. Following VerlTool [2], we train a deep-research agent equipped with Google search engine and a sandboxed Python executor as tools, with Qwen3-8B as the base model. We evaluate the performance on the General AI Assistants benchmark (GAIA), which tests LLM assistants' comprehensive capabilities in real world tasks requiring reasoning, web browsing, and tool-use proficiency.
>
>
> | Method | Level 1 | Level 2 | Level 3 | Avg |
> |---|---:|---:|---:|---:|
> | **Reasoning without Tool** |  |  |  |  |
> | Qwen3-32B-thinking | 26.2 | 12.1 | 0 | 14.9 |
> | DeepSeek-R1-32B | 21.5 | 13.6 | 0.0 | 14.2 |
> | QwQ-32B | 30.9 | 6.5 | 5.2 | 18.9 |
> | GPT-4o | 23.1 | 15.4 | 8.3 | 17.5 |
> | DeepSeek-R1-671B | 40.5 | 21.2 | 5.2 | 25.2 |
> | **Tool Integrated Reasoning (Qwen3-8B)** |  |  |  |  |
> | Vanilla RAG | 28.2 | 15.4 | 16.7 | 20.4 |
> | Search-o1 | 35.9 | 15.4 | 0.0 | 21.4 |
> | WebThinker | 43.6 | 11.5 | 0.0 | 22.3 |
> | ReAct | 35.9 | 17.3 | 8.3 | 23.3 |
> | *RL-based Method* |  |  |  |  |
> | Qwen3-8B | 28.1 | 15.4 | 16.7 | 20.4 |
> | + GRPO | 48.7 | 32.7 | 16.7 | 36.9 |
> | + Stratified GRPO | **61.5** | **44.2** | **33.3** | **49.5** |
>
> Stratified GRPO substantially improves over GRPO (**+12.6 avg**) and outperforms other strong baselines. Importantly, these gains arise in a domain **far beyond QA retrieval over fixed corpora**, but involving open-web search, tool invocation decisions, longer-horizon planning, and reasoning. This directly demonstrates Stratified GRPO’s broader applicability to tool-augmented reasoning agents. We will include full experimental details and analysis in the revision.
>
>
>
> **Q8 Lack of comparison with structure-aware critic baselines.**
>
> We respectfully posit that the "Structure-Aware PPO" suggested is not an existing baseline, but effectively an alternative implementation of our own conceptual contribution. The "Structure-Aware PPO" is **not an established prior method in the LLM search-agent literature**. To our knowledge, existing PPO-style LLM search agents do **not** condition critics on structural variables.
>
> More importantly, this suggestion is **conceptually aligned with our contribution**, not orthogonal to it. Our central claim is that _any_ policy-gradient method, with or without critics, that uses a baseline **not conditioned on the heterogeneity driver** will incur cross-stratum bias. We state this explicitly: Theorem 1’s discussion notes that PPO can also suffer the same structural bias when the critic ignores stratum structure. In this sense, a stratum-conditioned PPO critic is effectively **another implementation of our core idea** (handling trajectory heterogeneity by conditioning credit assignment on structure). Comparing against it would not be comparing against prior work, but rather comparing two implementations (GRPO vs. PPO) of _our proposed strategy_.
>
> Empirically, we already compare against **Search-R1**, a strong PPO-based search-agent baseline with a learned critic, and Stratified GRPO still outperforms it consistently. This supports our prediction that _critic learning alone, without structural conditioning_, does not resolve heterogeneity-induced bias. We agree that exploring conditional critics is promising future work. However, its absence should not be interpreted as omitting a standard prior baseline, since such structure-aware critics are not currently part of existing search-agent RL methods.
>
>
>
> [1] _Search-R1: Training LLMs to Reason and Leverage Search Engines with Reinforcement Learning._ COLM 2025.
> [2] _VerlTool: Towards Holistic Agentic Reinforcement Learning with Tool Use._ arXiv 2025.

---

> > ### Comment · Reviewer_ukKu · 2025-11-24
> > **Thanks for response**
> >
> > Thank you for your response. I have carefully read your reply. However, I find the experimental evidence in your work insufficiently convincing. If you wish to persuade me to raise my score, please include more compelling experimental results in the revised manuscript.
> >
> > For instance, your theoretical analysis claims that your method reduces variance—what empirical evidence supports this claim? Please provide complete curves for the following metrics, as previously listed in my questions:
> >
> > - Policy entropy over time,
> > - Gradient norm dynamics,
> > - Variance of importance weights,
> > - Distribution or average of rollout lengths,
> > - Variance of advantage estimates,
> > - KL divergence between the current and reference policies.
> >
> > Your current response did not directly address these specific concerns. Without clear and convincing empirical results demonstrating the claimed benefits—particularly the variance reduction—I will maintain my current assessment, and may even lower my score. **Like Reviewer cot2, I currently consider the theoretical contribution to be trivial and lacking in persuasiveness.**

---

> ### Author Response · Authors · 2025-11-24
> **Response 4**
>
> We thank the reviewer for the follow-up and appreciate the opportunity to clarify the points you raised.
>
> **Q9 Empirical Evidence for Stability**
>
> We realize our previous response **Q1** may not have highlighted this prominently: **we already provided the requested stability curves averaged across 3 independent runs (Gradient Norms, KL Divergence, Training Reward, and Search Calls) in Appendix D (Figure 2) of the revised PDF.**
>
> To fully address your request and eliminate any ambiguity, we further added **Rollout Length, Entropy, and Global Advantage Variance curves averaged across 3 independent runs in Appendix D (Figure 3) of the revised PDF**.
>
> Taken together, these plots provide consistent evidence that **Stratified GRPO yields substantially more stable optimization and improved exploration** than GRPO:
>
> 1.  **Gradient norms (Figure 2, col. 3):** GRPO produces larger and more erratic gradient norms, with pronounced spikes. In contrast, Stratified GRPO maintains lower and more controlled gradients, reflecting substantially more stable updates.
> 2.  **KL divergence (Figure  2, col. 4):** Stratified GRPO exhibits consistently lower KL divergence from the reference policy than GRPO, indicating a more stable and less brittle optimization trajectory.
> 3.  **Training reward (Figure  2, col. 1):** Stratified GRPO attains consistently higher rewards throughout training and yields significantly better evaluation performance than GRPO. The evaluation performance is shown in our response to **Q4**, averaged over three seeds.
> 4.  **Number of search calls (Figure  2, col. 2):** Stratified GRPO steadily increases the number of search calls over time, whereas GRPO stays near one call per question. This indicates that Stratified GRPO specifically encourages exploration of effective multi-turn search behaviors, rather than collapsing to a single-turn strategy.
> 5. **Variance of Importance Weights:** The importance weight is defined as $\rho_t = \frac{\pi_\theta(a_t|s_t)}{\pi_{\text{ref}}(a_t|s_t)}$. Its magnitude is tightly correlated with the divergence between the current and reference policies, which can be monitored by tracking the KL Divergence. Stratified GRPO consistently maintains **lower and smoother KL** than GRPO (Figure 2, col. 4), which provides strong evidence that the policy updates remain well-constrained and that the importance weights do not suffer from the explosive variance that characterizes unstable optimization.
> 6. **Rollout Length (Figure  3, row. 1 col. 1):**  Stratified GRPO successfully expands the average rollout length, whereas GRPO collapses to short responses. This aligns with the "Search Calls" plot, confirming that our method encourages long-horizon multi-turn search exploration.
> 7. **Entropy (Figure  3, row. 1 col. 2):** The entropy plot shows that Stratified GRPO's entropy converges at a similar rate to GRPO, but to a much higher-reward, longer-horizon policy.
> 8. **Global normalized advantage variance (Figure  3, row. 1 col. 3) and link to Theorems:** Theorems 1–2 analyze global variance **before normalization** and show that GRPO’s estimator contains a within-stratum term plus a cross-stratum component, and Stratified GRPO removes the latter. Both GRPO and Stratified GRPO normalize advantages. Therefore **both GRPO and Stratified GRPO have unit _global_ variance after normalization** (Theorem 5). Therefore, we do **not** claim that our final algorithm, Stratified GRPO, which normalizes advantages, exhibits a lower global normalized variance than GRPO; a flat unit-variance curve is expected for both and serves as a sanity check. The key theoretical distinction is **conditional behavior**: Stratified GRPO guarantees **zero mean and unit variance _within each stratum_** (Theorem 4), whereas GRPO has stratum-dependent bias and inconsistent within-stratum scaling. The practical consequences predicted by this conditional correction are exactly what Figures 2–3 show across seeds: smoother gradient norms, lower/smoother KL, and sustained growth in rollout length / multi-call search exploration.
>
> Finally, the multi-seed evaluation in **Q4**  shows substantial performance gains far exceeding the standard deviations across runs, confirming that the improvements are robust and not due to random seed selection. The hyperparameter sensitivity analyses in **Q5** further demonstrate the stability of our method. The broad high-performance region exists between $\alpha=0.4$ and $\alpha=0.8$, indicating the method does not require delicate tuning.

---

> > ### Author Response · Authors · 2025-11-24
> > **Response 5**
> >
> > **Q10 On the claim that the theoretical analysis is trivial or unpersuasive**
> >
> > We respectfully disagree that the theory is trivial. Our theoretical contribution is not the introduction of heavy math tools, but the **identification and formalization of a structural failure mode (cross-stratum bias)** in existing RL methods for training LLM search agents, together with **a principled correction via stratification.** This insight is supported by consistent empirical gains on challenging benchmarks (e.g., **+12.6 on GAIA for deep-research agents**, and **+11.3 points in the main-paper experiments**), indicating that this theoretical contribution is nontrivial and practically consequential.
> >
> > For clarity, we restate the core conceptual points:
> >
> > **Our intent and contribution.**
> > Our goal is not to introduce mathematically sophisticated tools, but to *identify and formalize* a structural issue—**cross-stratum bias**—that, to our knowledge, has not been analyzed in prior RL work for LLM search agents. Existing algorithms almost universally adopt a global baseline (e.g., GRPO, RLOO, PPO-style RLVR), implicitly assuming homogeneous trajectories. By decomposing the variance and moments of global versus stratified estimators (Thms. 1–5), we show that this assumption is systematically violated once trajectories differ in tool-calling structure, and that this directly affects exploration and stability.
> >
> > A key and non-obvious conceptual message is that *estimators with identical global moments (Theorem 5) can behave very differently once conditioned on strata (Theorem 4)*. This distinction turns out to be crucial for LLM search agents, and Table 1 indicates that several strong baselines still suffer from this structural issue.
> >
> > **Why the theory “looks simple” but is not trivial.**
> > Although we intentionally present the algebra in an accessible form, the underlying reasoning is not elementary. The proofs require handling non-trivial aspects of heterogeneous RL:
> >
> > 1) **$\theta$-dependent partitions:**
> >    Stratum membership $S(\tau;\theta)$ changes with the policy, requiring careful treatment of *moving* mixture measures $p_k(\theta)$ (Theorem 3).
> >
> > 2) **Hierarchical variance decomposition:**
> >    Extracting the structural bias term
> > $$\sum\_{k} n\_{k} \left(\overline{R}\_{k} - \overline{R}\_{\text{global}}\right)^2$$
> >    requires a mixture-model interpretation that goes beyond simple algebra (Theorems 1–2).
> >
> > 3) **Conditional vs. global consistency:**
> >    Showing that local (per-stratum) normalization still preserves global unbiasedness is subtle and requires careful conditioning arguments (Theorems 4–5).
> >
> > The real complexity lies in modeling shifting policy support and hierarchical reward structures—*not* in the final symbolic steps, which we intentionally simplify for clarity.

---

> > > ### Comment · Reviewer_ukKu · 2025-11-25
> > > **Thanks for your response**
> > >
> > > The advantage (adv) curves you presented are normalized—**after normalization, the variances of GRPO and Stratified GRPO appear identical** (Figure 3c). This suggests that, during training, **the two methods exhibit the same variance in their advantage estimates**. Consequently, the reason why Stratified GRPO outperforms GRPO becomes unclear—at the very least, it contradicts your theoretical claim (at least the performance improment is not raised from variance reduction).
> > >
> > > Since your theoretical analysis claims that the **variance can be reduced**, why don’t you provide more compelling experimental evidence?   So, I’m even more confident now that my judgment is correct.

---

> > > > ### Comment · Reviewer_ukKu · 2025-11-26
> > > > **I have another question about the rollout length**
> > > >
> > > > Figure 3(a) shows that the rollout length of GRPO continuously decreases, while Figure 2(a) shows its reward steadily increasing. Have you examined the rollout trajectories to identify any patterns that might explain this phenomenon?
> > > >
> > > >  Lines 1109–1110 state: “with a maximum response length and retrieved content length of 500 tokens per interaction turn.” Could this length constraint be contributing to the observed behavior?
> > > >
> > > >  In my own experiments, I’ve found that when the response length is set too short—or when the model frequently exceeds the length limit and consequently fails to receive positive rewards—it tends to produce progressively shorter outputs over time.

---

> > > > > ### Author Response · Authors · 2025-11-26
> > > > > **Response 7**
> > > > >
> > > > > We thank the reviewer for the additional question.
> > > > >
> > > > > **Q12 The rollout length of GRPO continuously decreases, while its reward is increasing?**
> > > > >
> > > > > We respectfully note that our method is built on the established LLM search-agent RL of Search-R1's official codebase [1], and we follow their official setups. In the public wandb logs released with the Search-R1 github repository, the runs exhibit **the same pattern** as our observation in Figure 2 and Figure 3 that GRPO's rollout length decreases while the reward increases during training. Thus, the behavior we observe for GRPO is consistent with prior work and appears to be a characteristic of the baseline itself rather than specific to our implementation.
> > > > >
> > > > > **Q13 Length limit and truncation**
> > > > >
> > > > > The 500-token limit applies to the retrieved (searched) document per interaction turn, **not** to the model’s response generation. The maximum model generation length is 4096 tokens. **This configuration follows the official Search-R1 implementation exactly.** As described in Section B.2 (“Experimental Settings”) of Search-R1, they use the same retrieval budget and response length settings, and we adopt their publicly released training code without changing these values to ensure strict comparability of our GRPO baseline.
> > > > >
> > > > > Moreover, in our rollout length plot, the average rollout lengths for both GRPO and Stratified GRPO remain far below the 4096-token maximum, so generation rarely approach the cap and get truncation. For this reason, we do not believe length truncation is driving the observed decrease in rollout length for GRPO.
> > > > >
> > > > >
> > > > > [1] _Search-R1: Training LLMs to Reason and Leverage Search Engines with Reinforcement Learning._ COLM 2025.

---

> > > > > > ### Comment · Reviewer_ukKu · 2025-11-26
> > > > > > **Will you release a complete and runnable codebase during the rebuttal period?**
> > > > > >
> > > > > > Thanks again for your response.
> > > > > >
> > > > > > Did you use normalized advantages during training? If so, wouldn't the impact of advantage variance on training be effectively eliminated? Conversely, if you did **not** use normalized advantages for training, why did you plot normalized advantages in your figures (e.g., Figure 3 in the last version)?
> > > > > >
> > > > > > Given this inconsistency, I’d like to reproduce your results myself to verify my understanding. Your model is relatively small, and the number of training steps is modest. I should be able to run the full experiment fairly quickly.
> > > > > >
> > > > > > Clarification: I’m not pressuring you to release your code. It’s just that certain aspects of the rebuttal process have raised doubts for me, so I believe it would be more reliable to make my final decision only after rigorously verifying the results myself.

---

> > > > > > > ### Author Response · Authors · 2025-11-26
> > > > > > > **Response 8**
> > > > > > >
> > > > > > > We thank the reviewer for the continued engagement and would like to clarify a few points where there still seems to be a misunderstanding of what is – and is not – claimed in the paper.
> > > > > > >
> > > > > > > **Q14 On normalized vs. unnormalized advantages and Figure 3**
> > > > > > >
> > > > > > > **1. What we claim (and do **not** claim) about variance**
> > > > > > >
> > > > > > > Our central claim is **not** that “variance is reduced and therefore training improves.”
> > > > > > >
> > > > > > > What we emphasize is:
> > > > > > >
> > > > > > > -   A _structural_ flaw of global baselines/normalization in the presence of heterogeneous strata (cross-stratum bias); and
> > > > > > >
> > > > > > > -   A key **conditional property** of  Stratified GRPO: within each stratum, it provides a **zero-mean, unit-variance globally and within each stratum**, whereas standard GRPO produces **biased and stratum-dependent variance** within each stratum.
> > > > > > >
> > > > > > > This is summarized by Theorem 4 and Theorem 5 in the paper:
> > > > > > >
> > > > > > > -   **Globally**, both Stratified GRPO and GRPO have mean 0 and variance 1.
> > > > > > >
> > > > > > > -   **Conditionally within each stratum**, Stratified GRPO is unbiased and unit-variance, while GRPO’s mean and variance depend on that stratum’s reward statistics.
> > > > > > >
> > > > > > > **The main advantage of Stratified GRPO that we claim is precisely this _per-stratum_ purity and stability of the signal.**
> > > > > > >
> > > > > > > We also analyze variance decompositions for **unnormalized** global and stratified advantages (Theorem 1 and Theorem 2) to make the structural effect of cross-stratum bias explicit. These results apply **only to unnormalized advantages**; we do **not** claim them as properties of the final Stratified GRPO algorithm.
> > > > > > >
> > > > > > >
> > > > > > > 2.  **What is implemented in the code?**
> > > > > > >
> > > > > > > We use the official GRPO implementation in Verl, which uses **globally normalized advantages**. For Stratified GRPO, the only change is to normalize **within each stratum** rather than globally (Algorithm 1). This is exactly what is described in Sections 3.3–4; there is no discrepancy between the algorithm and the implementation.
> > > > > > >
> > > > > > > 4.  **What is plotted in Figure 3?**
> > > > > > > Earlier in the discussion, you asked for additional diagnostics on stability and variance. In our response to Q11, we updated Figure 3 to report:
> > > > > > >
> > > > > > > -   The **unnormalized global** advantage variance; and
> > > > > > >
> > > > > > > -   The **global** and **in-stratum** advantage variance **after normalization**.
> > > > > > >
> > > > > > >
> > > > > > > The updated figure shows that:
> > > > > > >
> > > > > > > -   For **unnormalized global advantages**, Stratified GRPO consistently reduces variance relative to GRPO, matching Theorem 1.
> > > > > > >
> > > > > > > -   For **normalized advantages**, the **global** variance is close to 1 for both methods, but the **conditional in-stratum** behavior differs: Stratified GRPO maintains in-stratum unit variance, while GRPO does not (as analyzed in Theorems 4 and 5).
> > > > > > >
> > > > > > >
> > > > > > > **In short, there is no inconsistency**:
> > > > > > > The theory first analyzes unnormalized advantages to expose cross-stratum bias, then shows how normalization interacts with that structure; the implementation and updated plots follow this analysis.

---

> > > > > > > ### Author Response · Authors · 2025-11-26
> > > > > > > **Response 9**
> > > > > > >
> > > > > > > **Q15 On reproducing the results and code release**
> > > > > > >
> > > > > > > We welcome independent attempts to reproduce our results, and have added a reproducibility statement in Section F of the Appendix to offer practical guidance.
> > > > > > >
> > > > > > > For **factual QA tasks**, we follow the settings of Search-R1 and build on their **official** codebase.
> > > > > > > For **deep-research agent tasks**, we follow the settings of VerlTool and build on their **official** codebase.
> > > > > > >
> > > > > > > Both codebases use Verl for RL training.
> > > > > > >
> > > > > > > Our method is a **minimal modification** of the official GRPO implementation in Verl: the core change is to group rollouts by **(prompt, search-count/tool-calling-count)** and apply **per-stratum normalization**.
> > > > > > >
> > > > > > > For example, in VERL’s official function for computing the GRPO advantage, rollouts in the batch are grouped into each prompt’s sampled rollouts based on the prompt index:
> > > > > > >
> > > > > > > ```
> > > > > > > #VERL GRPO advantage
> > > > > > > def compute_grpo_outcome_advantage(token_level_rewards: torch.Tensor,
> > > > > > >                                    eos_mask: torch.Tensor,
> > > > > > >                                    index: torch.Tensor,
> > > > > > >                                    epsilon: float = 1e-6):
> > > > > > >
> > > > > > >     response_length = token_level_rewards.shape[-1]
> > > > > > >     non_zero_mask = (token_level_rewards != 0)
> > > > > > >     scores = (token_level_rewards * non_zero_mask).sum(dim=-1)
> > > > > > >
> > > > > > >     id2score = defaultdict(list)
> > > > > > >     id2mean = {}
> > > > > > >     id2std = {}
> > > > > > >
> > > > > > >     with torch.no_grad():
> > > > > > >         bsz = scores.shape[0]
> > > > > > >         for i in range(bsz):
> > > > > > >             id2score[index[i]].append(scores[i])
> > > > > > >
> > > > > > >     # ... compute per-prompt mean/std and normalized advantages
> > > > > > > ```
> > > > > > >
> > > > > > > In our **stratified** version, the only modification is to extend the grouping key with the search-count information, i.e., replacing
> > > > > > >
> > > > > > > `id2score[index[i]].append(scores[i])`
> > > > > > >
> > > > > > > with
> > > > > > >
> > > > > > > `id2score[(int(index[i]), int(search_counts[i]))].append(scores[i])`
> > > > > > >
> > > > > > > while keeping the remaining identical. This change is minimal and should be straightforward to reproduce.
> > > > > > >
> > > > > > > We provide full algorithmic details, hyperparameters, model sizes, training steps, and evaluation protocols in the paper and appendix, precisely so that a practitioner can re-implement the method straightforwardly.
> > > > > > >
> > > > > > > -   **The algorithm is fully specified in the paper.**
> > > > > > >     Algorithm 1 provides a step-by-step description of Stratified GRPO, including the computation of global statistics, per-stratum statistics, SAN, and the blended advantage used in the policy gradient update. No new networks, optimizers, or training tricks beyond standard GRPO/RLVR are introduced, so an experienced practitioner can directly port these steps into an existing codebase.
> > > > > > >
> > > > > > > -   **Design choices and hyperparameters are completely documented.**
> > > > > > >     Sections 3–5 and Appendix E detail the stratification strategy, model choices, and all training hyperparameters.
> > > > > > >
> > > > > > > -   **Code release plan.**
> > > > > > >     In line with ICLR policy and standard practice, we are committed to releasing an implementation of Stratified GRPO and trained model checkpoints upon acceptance, which will make the small set of required changes even more transparent to the community.

---

> > > > > > > > ### Comment · Reviewer_ukKu · 2025-11-26
> > > > > > > > **Thanks for your response and releasing the core code.**
> > > > > > > >
> > > > > > > > I will try to reproduce it on VERL.

---

> > > > > > > > > ### Author Response · Authors · 2025-11-27
> > > > > > > > > **Response 10**
> > > > > > > > >
> > > > > > > > > Thank you for your willingness to try reproducing our results. Since this is a large-scale LLM RL setup and a single run can take one or more days depending on hardware, we would like to share a few practical notes that may help ensure your implementation matches ours:
> > > > > > > > >
> > > > > > > > > **1. Settings, environment, and codebase:** Stratified GRPO is implemented on the official Search-R1 codebase with their official environment. We use a global batch size of 256, a mini-batch size of 256, and a learning rate of 1e-6 with a warm-up ratio of 0.1. Training is conducted for 200 steps with Exact Match used as the training reward. Other hyparameter settings are listed in detail in Appendix E.
> > > > > > > > >
> > > > > > > > > **2. Early Sanity-Check:** To help you verify the run is proceeding correctly without waiting for full training (which takes one or more days depending on devices), a correct implementation typically shows clear qualitative trends early in training (around **Step 50**):
> > > > > > > > >
> > > > > > > > > -   **Rollout Length (Figure 3, row 1 col 1):**
> > > > > > > > >
> > > > > > > > >     -   **Baseline GRPO:** The rollout length should decrease and stabilize for baseline GRPO.
> > > > > > > > >
> > > > > > > > >     -   **Stratified GRPO:** The rollout length should increase for Stratified  GRPO.
> > > > > > > > >
> > > > > > > > > -   **Search Calls (Figure 2, col 2):**
> > > > > > > > >
> > > > > > > > > 	   -   **Baseline GRPO:**  Baseline GRPO should have number of search calls stays around 1.
> > > > > > > > >
> > > > > > > > > 	   -   **Stratified GRPO:** Stratified GRPO should show a gradual increase in the number of search calls.
> > > > > > > > >
> > > > > > > > > These baseline GRPO trends are consistent with those reported for Search-R1 and can be observed in their official logs in their github repository.
> > > > > > > > >
> > > > > > > > > If any aspect of the setup is unclear, we are very happy to clarify further. We appreciate your effort to engage with the method at this level of detail.

---

> > > > ### Author Response · Authors · 2025-11-26
> > > > **Response 6**
> > > >
> > > > We thank the reviewer for the follow-up. In the revised manuscript we have updated **Figure 3** to report the full set of advantage variance metrics.
> > > >
> > > > **Q11 Global normalized advantage curves appear identical**
> > > >
> > > > Our theoretical variance reduction result (Theorem 1) concerns **the variance of the _unnormalized_ global advantages**. In the updated Figure 3 (row 2, col 2), we now plot this quantity and observe that Stratified GRPO consistently exhibits lower unnormalized global advantage variance than GRPO throughout training, in line with Theorem 1.
> > > >
> > > > The performance gains of Stratified GRPO, however, are primarily due to its **per-stratum behavior**. As shown in Theorem 4, Stratified GRPO is designed so that the _normalized_ advantages are zero-mean and unit-variance **within each stratum**, whereas GRPO’s global normalization cannot guarantee this property. The updated Figure 3 (row 2, col 1) empirically validates this: per-stratum normalized variances for Stratified GRPO stay at 1, while GRPO cannot. Furthermore, Stratified GRPO's in-stratum normalized advantage variance is consistently lower than GRPO, further demonstrating our method's stability.
> > > >
> > > > Finally, Theorem 5 states that, after normalization, **both** GRPO and Stratified GRPO have unit _global_ variance. Thus, the near-identical global normalized variance curves in Figure 3 (row 1, col 3) are expected and consistent with our theory; they do not contradict our variance reduction claim, which applies to the unnormalized statistics only. Overall, all these empircal metrics align well with our theory.

---

### Official Review · Reviewer_cot2 · 2025-10-15

**Soundness:** 3
**Presentation:** 2
**Contribution:** 2
**Rating:** 4
**Confidence:** 4

**Summary:**

This paper introduces the Stratified Advantage Normalization (SAN) technique, which aims to eliminate the cross-stratum bias in the GRPO algorithm. By combining SAN with the vanilla Global Normalized (GN) technique through weighted average, the authors propose the Stratified GRPO algorithm. They further provide detailed theoretical analyses and demonstrate performance improvements on several question-answering benchmarks. Overall, I don’t find this paper appealing.

**Strengths:**

- The paper has a clear motivation and is well structured.
- The proposed algorithm shows empirical improvements (and also looks strong).

**Weaknesses:**

- The theoretical analysis is trivial, I believe even a high school student could understand it.
- The authors devote nearly five pages to discussing a rather trivial theoretical analysis, which feels somewhat redundant.
- The baseline algorithms are limited.
- Lacks sensitivity analysis of $\alpha\in[0,1]$.

**Questions:**

- The theoretical analysis is overly redundant, and the results are trivial. In contrast, the experimental section appears insufficient. I suggest balancing the length between the theoretical analysis and the experimental section.
- The main part of the experiments consists of Table 1 (main experiments) and Table 2 (ablation studies). I believe additional discussion on the sensitivity of the important hyperparameter $\alpha$ (beyond just zero and one) is needed to enrich the experimental section. In the Appendix D.1, the $\alpha$ value for the Qwen 2.5 3B Base model is set to $0.6$, which is already close to taking the average of SAN and GN.
- The authors divide different strata based on search count. I believe they should add an additional baseline where the strata are randomly assigned, in order to rigorously demonstrate the effectiveness of the proposed algorithm. I think this experiment is important and encourage the authors to include it.

---

> ### Author Response · Authors · 2025-11-22
> **Response 1**
>
> **Q1. Theoretical analysis is trivial and redundant**
>
> **Our intent and contribution.**
> Our goal is not to introduce mathematically sophisticated tools, but to *identify and formalize* a structural issue—**cross-stratum bias**—that, to our knowledge, has not been analyzed in prior RL work for LLM search agents. Existing algorithms almost universally adopt a global baseline (e.g., GRPO, RLOO, PPO-style RLVR), implicitly assuming homogeneous trajectories. By decomposing the variance and moments of global versus stratified estimators (Thms. 1–5), we show that this assumption is systematically violated once trajectories differ in tool-calling structure, and that this directly affects exploration and stability.
>
> A key and non-obvious conceptual message is that *estimators with identical global moments (Theorem 5) can behave very differently once conditioned on strata (Theorem 4)*. This distinction turns out to be crucial for LLM search agents, and Table 1 indicates that several strong baselines still suffer from this structural issue.
>
> **Why the theory “looks simple” but is not trivial.**
> Although we intentionally present the algebra in an accessible form, the underlying reasoning is not elementary. The proofs require handling non-trivial aspects of heterogeneous RL:
>
> 1. **$\theta$-dependent partitions:**
>    Stratum membership $S(\tau;\theta)$ changes with the policy, requiring careful treatment of *moving* mixture measures $p_k(\theta)$ (Theorem 3).
>
> 2. **Hierarchical variance decomposition:**
>    Extracting the structural bias term
> $$\sum\_{k} n\_{k} \left(\overline{R}\_{k} - \overline{R}\_{\text{global}}\right)^2$$
>    requires a mixture-model interpretation that goes beyond simple algebra (Theorems 1–2).
>
> 3. **Conditional vs. global consistency:**
>    Showing that local (per-stratum) normalization still preserves global unbiasedness is subtle and requires careful conditioning arguments (Theorems 4–5).
>
> The real complexity lies in modeling shifting policy support and hierarchical reward structures—*not* in the final symbolic steps, which we intentionally simplify for clarity.
>
> **Revision to address redundancy.**
> We agree that some parts of Section 2 can be streamlined. In the revision, we will move several proof sketches to the appendix and keep only the key identities and take-away messages in the main text. These changes will reduce the perceived redundancy while keeping the conceptual contribution intact.
>
>
>
> **Q2 The experiment section appears insufficient.**
>
> We agree that broader validation is valuable. To further evaluate generality, we add experiments on **tool-use agents in real-world multi-step settings**. Following VerlTool [1], we train a deep-research agent equipped with Google search engine and a sandboxed Python executor as tools. We evaluate the performance on the General AI Assistants benchmark (GAIA), which tests LLM assistants' comprehensive capabilities in real-world tasks requiring reasoning, web browsing, and tool-use proficiency.
>
>
> | Method | Level 1 | Level 2 | Level 3 | Avg |
> |---|---:|---:|---:|---:|
> | **Reasoning without Tool** |  |  |  |  |
> | Qwen3-32B-thinking | 26.2 | 12.1 | 0 | 14.9 |
> | DeepSeek-R1-32B | 21.5 | 13.6 | 0.0 | 14.2 |
> | QwQ-32B | 30.9 | 6.5 | 5.2 | 18.9 |
> | GPT-4o | 23.1 | 15.4 | 8.3 | 17.5 |
> | DeepSeek-R1-671B | 40.5 | 21.2 | 5.2 | 25.2 |
> | **Tool Integrated Reasoning (Qwen3-8B)** |  |  |  |  |
> | Vanilla RAG | 28.2 | 15.4 | 16.7 | 20.4 |
> | Search-o1 | 35.9 | 15.4 | 0.0 | 21.4 |
> | WebThinker | 43.6 | 11.5 | 0.0 | 22.3 |
> | ReAct | 35.9 | 17.3 | 8.3 | 23.3 |
> | *RL-based Method* |  |  |  |  |
> | Qwen3-8B | 28.1 | 15.4 | 16.7 | 20.4 |
> | + GRPO | 48.7 | 32.7 | 16.7 | 36.9 |
> | + Stratified GRPO | **61.5** | **44.2** | **33.3** | **49.5** |
>
> Stratified GRPO improves over GRPO by **+12.6 Avg** and exceeds other strong baselines. Importantly, these gains arise in a setting that requires **open-web search, tool-invocation decisions, longer-horizon planning, and multi-step reasoning**, far beyond fixed-corpus QA. This directly supports the broader applicability of our approach. Full experiment details and analysis will be included in the revision.
>
> [1] _VerlTool: Towards Holistic Agentic Reinforcement Learning with Tool Use._ arXiv 2025.

---

> ### Author Response · Authors · 2025-11-22
> **Response 2**
>
> **Q3 Baseline where the strata are randomly assigned**
>
> We appreciate this suggestion and agree it is a clean sanity check. Importantly, our theory predicts that **random stratification should not help in expectation**: when strata are assigned randomly, stratum means coincide with the global mean up to sampling noise, so Theorem 1 implies no systematic variance reduction or bias removal (equality case).
>
> We add this experiment of randomly assigning each prompt's sampled rollouts to 2 strata or 3 strata, with all other hyperparameters aligning with our methods. The following table shows that Random strata assignment leads to performance similar to GRPO, which is far inferior than our Stratified GRPO.
>
> |  Method  | NQ | TriviaQA | PopQA | HotpotQA | 2Wiki | Musique | Bamboogle |Avg  |
> |---|---:|---:|---:|---:|---:|---:|---:|---:|
> | GRPO | 45.2 | 61.2 | 43.8 | 32.6 | 29.7 | 7.8| 12.9 | 33.3 |
> | randomly assign 2 strata | 43.9 | 60.7 | 43.9 | 31.6 | 29.4 | 7.8 | 14.5 | 33.1 |
> | randomly assign 3 strata | 44.1 | 61.4 | 43.6 | 32.5 | 28.0 | 7.7 | 15.3 | 33.2|
> | Stratified GRPO | 45.9 | 61.4 | 43.0 | 40.8 | 39.9 | 17.7 | 42.7 | 41.6 |
>
> As predicted, random stratification yields performance almost identical to GRPO (33.1–33.2 vs. 33.3 avg), while **Stratified GRPO improves the average by +8.3 points**, with particularly large gains on multi-hop datasets. This directly supports our claim that the benefit comes from **structure-aware** stratification, not simply from partitioning trajectories.
>
> **Q4 The baseline algorithms are limited.**
>
> We respectfully disagree. Our evaluation already covers a **broad and relevant set of strong baselines**, including:
>
> -   **Non-RL:** Direct Generation, SFT, RAG, Search-o1, IRCoT
>
> -   **RL with search:** **Search-R1 (PPO-based)**, **ReSearch**, **GRPO**
>
> -   **RL without search:** R1
>
>
> These baselines span (1) prompting/SFT/RAG, (2) RLVR with and without tools using PPO or GRPO.  In particular, Search-R1 provides a strong PPO-based comparator, and GRPO reflects the most common baseline choice for LLM-agent RL. We therefore believe the baseline suite is comprehensive for this setting.
>
> **Q5 Sensitivity analysis of $\alpha$; $\alpha=0.6$ seems close to a simple average of SAN and GN.**
>
> We appreciate this suggestion  of sensitivity analysis of $\alpha$. We now provide a detailed sweep of $\alpha \in \\{0.0, 0.2, 0.4, 0.6, 0.8, 1.0\\}$ on Qwen-2.5-3B-Base:
>
> |  value of $\alpha$  | NQ | TriviaQA | PopQA | HotpotQA | 2Wiki | Musique | Bamboogle |Avg  |
> |---|---:|---:|---:|---:|---:|---:|---:|---:|
> | 0.0 (GRPO) | 45.2 | 61.2 | 43.8 | 32.6 | 29.7 | 7.8| 12.9 | 33.3 |
> | 0.2 | 45.1 | 61.8 | 43.1 | 32.2 | 29.5 | 7.6 | 14.5 | 33.4 |
> | 0.4 | 44.6 | 59.8 | 43.2 | 39.6 | 36.5 | 15.6 | 39.5 | 39.8 |
> | 0.6 | 45.9 | 61.4 | 43.0 | 40.8 | 39.9 | 17.7 | 42.7 | 41.6 |
> | 0.8 | 45.1 | 60.1 | 43.9 | 40.3 | 41.9 | 17.5 | 37.9 | 41.0 |
> | 1.0 (SAN-only) | 43.7 | 59.3 | 41.1 | 36.6 | 38.4 | 12.6 | 25.0 | 36.7 |
>
>
> Key observations:
>
> -   **Robust improvement over GRPO.** All $\alpha \ge 0.4$ significantly outperform GRPO, with a broad plateau of strong performance between 0.4 and 0.8. This shows that the method is robust to the exact choice of $\alpha$.
>
> -   **SAN-only is already strong but blending helps.** $\alpha = 1.0$ (SAN-only) still improves substantially over GRPO on average and especially on multi-hop datasets, but advantage blending yields the best overall trade-off, as our theory suggests.
>
>
> Regarding the “simple average” remark: note that $\alpha$ interpolates between **normalized** advantage signals whose scales are shaped by per-stratum vs. global statistics; an $\alpha$ of 0.6 is not equivalent to averaging raw rewards or baselines. The empirical sweep also shows that performance is not symmetric around 0.5, further confirming that $\alpha$ is not just a cosmetic averaging weight.

---

> > ### Comment · Reviewer_cot2 · 2025-11-25
> >
> > Thank you for your response. I believe the current PDF version does need to better balance the theoretical and experimental parts, especially since the experiments section only begins on page 7. The authors have added sensitivity analyses as well as randomly assigning 2 and 3 strata settings, but I think these changes require substantial revisions to the current paper.
> >
> > The theoretical part is indeed rather trivial to me. Empirically, the paper essentially introduces a new normalization method, SAN, which feels quite unremarkable. The performance improvements are also fairly expected in my view.
> >
> > Therefore, I will maintain my score and encourage the authors to participate in a new round of review.

---

> > > ### Author Response · Authors · 2025-11-26
> > > **Response 3**
> > >
> > > Thank you again for your careful reading of the paper and for your additional comment. We would like to clarify how the current revision already addresses the remaining concerns you raised and why we believe the work is ready for acceptance rather than requiring a new round of review.
> > >
> > > **Q6 Balance between theory and experiments**
> > >
> > > We agree that the initial version was theory-heavy. In the new PDF we have:
> > >
> > > -   Streamlined the theoretical exposition by moving proof sketches and auxiliary lemmas to the appendix and focusing the main text on the key ideas and take-aways.
> > >
> > > -   Expanded the empirical section to include:
> > >
> > >     -   seven QA benchmarks with a broad set of non-RL and RL baselines (Table 1),
> > >
> > >     -   the GAIA deep-research benchmark with tool-use agents (Table 2),
> > >
> > >     -   an ablation study (Table 3),
> > >
> > >     -   a sensitivity analysis of the blending coefficient $\alpha$ (Table 4), and
> > >
> > >     -   training-dynamics analyses (Figure 1) showing reward stability and learned search policies.
> > >
> > > Additional experiments requested during the review: **random-strata controls, multi-seed evaluation, and full training-dynamics analyses**, are included in **Appendix D**. We kept these in the appendix to respect the 10-page limit while still presenting all requested evidence.
> > >
> > > We therefore believe the present balance between theory and experiments is appropriate and does not require further structural revision.
> > >
> > > We would also like to clarify that the **main experiment protocol in the original submission already followed standard practice for LLM search agent RL** like Search-R1 [1,2,3,4,5]. The additional experiments now included: deep-research agent tasks, extended ablations, random-strata controls, and a detailed α sensitivity sweep, were added in direct response to your helpful suggestions. They go beyond what is typically required to validate a new RL method for LLM search agent and should therefore be viewed as **supplementary evidence that further strengthens**, rather than _enables_, our main claims.
> > >
> > > **Q7 On the “triviality” of the theory and of SAN**
> > >
> > >
> > > We respectfully disagree that _the theoretical component is trivial or that SAN is merely an unremarkable normalization tweak_.
> > >
> > > 1.  **Problem formulation.** To our best knowledge, **this is the first work** to explicitly identify _cross-stratum bias_ in RL for LLM search agents and to show that using a global baseline over structurally heterogeneous trajectories (different numbers of tool calls) introduces a systematic, structure-dependent offset in the advantage. Proposition 1 and Theorem 1 formalize this bias and its precise contribution to variance, directly characterizing widely used RL methods such as GRPO, RLOO, and PPO with a critic that ignores the stratum.
> > >
> > > 2.  **Design and analysis of SAN.** SAN is motivated by this structural diagnosis: it is not “just” another baseline, but a per-stratum estimator with (i) invariance to positive affine transformations of the reward (Proposition 2), (ii) an exact variance decomposition relative to global normalization (Theorem 2), and (iii) a population-gradient characterization (Theorem 3) showing that SAN optimizes a weighted sum of _true within-stratum_ policy gradients. These results explain _why_ SAN should improve exploration and stability when trajectory structure matters.
> > >
> > > 3.  **Conditional vs global properties.** Theorem 4 shows that SAN is conditionally unbiased with unit variance within each stratum, whereas global normalization is conditionally biased and inconsistently scaled, even though both share identical global first and second moments (Theorem 5). This highlights a structural flaw of the widely adopted global normalization that is not captured by standard variance arguments.
> > >
> > > We fully acknowledge that the algebra itself is intentionally kept accessible; however, in our view the novelty lies in (i) identifying the right structural notion of heterogeneity for search agents, and (ii) connecting this to concrete algorithmic design and conditional-moment analysis. Many impactful ICLR papers introduce conceptually simple but carefully analyzed modifications; our work fits squarely in that tradition.
> > >
> > >
> > > [1] Search-R1: Training LLMs to Reason and Leverage Search Engines with Reinforcement Learning. COLM 2025.
> > >
> > > [2] StepSearch: Igniting LLMs Search Ability via Step-Wise Proximal Policy Optimization. EMNLP 2025.
> > >
> > > [3] ZEROSEARCH: Incentivize the Search Capability of LLMs without Searching. Arxiv 2025.
> > >
> > > [4] O2 -Searcher: A Searching-based Agent Model for Open-Domain Open-Ended Question Answering. Arxiv 2025.
> > >
> > > [5] ReSearch: Learning to Reason with Search for LLMs via Reinforcement Learning. Arxiv 2025.

---

> > > ### Author Response · Authors · 2025-11-26
> > > **Response 4**
> > >
> > > **Q8 On the empirical significance and whether gains are “expected”**
> > >
> > > We also respectfully disagree that the empirical improvements are merely expected or marginal.
> > >
> > > -   On factual QA, **Stratified GRPO consistently outperforms all baselines across seven datasets**. From Table 1, it improves upon GRPO by up to **11.3 points on average**, and yields **up to 14.5-point gains on multi-hop benchmarks**, where structural search behavior matters most.
> > >
> > > -   On GAIA (Table 2), **Stratified GRPO improves substantially over GRPO across Level 1–3 tasks by 12.6 points on average**, and on the hardest Level-3 questions it achieves about a **2× relative improvement over GRPO (33.3 vs. 16.7)**. These are challenging open-web, tool-use, and long-horizon tasks, and to our knowledge no prior RL method has achieved comparable improvements from the same base models.
> > >
> > > -   The **ablation study** (Table 3) shows that SAN alone already provides clear gains over GRPO, and that adding blending further boosts performance, especially on multi-hop QA. This directly links the theoretical design choices to empirical outcomes.
> > >
> > > -   The **$\alpha$ sensitivity analysis** (Table 4) demonstrates a broad performance plateau for $\alpha$  ∈ [0.4, 0.8], indicating that the method is robust rather than overly sensitive to tuning. Even the SAN-only variant ($\alpha$  = 1.0) remains substantially better than GRPO.
> > >
> > > -   The **training-dynamics plots** (Figure 1, 2, 3) show that, compared to GRPO, Stratified GRPO (i) avoids the reward-collapse issue on the instruct model and maintains a stable, monotonically increasing reward signal, and (ii) learns a multi-turn search policy that converges to about **2.5 search calls per question**, whereas GRPO stays near a single call. This connects the structural analysis of cross-stratum bias directly to better exploration and more effective multi-step search behavior.
> > >
> > > Taken together, these results go well beyond what we would expect from a generic normalization tweak and point to a genuinely useful structural improvement for training search agents.
> > >
> > >
> > > ---
> > >
> > > All specific points raised in your initial review, additional baselines, sensitivity analysis of α, random-strata control experiments, clearer exposition, and more extensive experiments, have already been implemented in the current PDF. Given the combination of (1) a clear and general structural diagnosis, (2) a principled and thoroughly analyzed estimator, and (3) strong, robust empirical gains across challenging benchmarks, we respectfully ask the reviewers to reconsider the overall assessment of the paper.

---

> ### Author Response · Authors · 2025-11-26
>
> Dear Reviewer cot2,
>
> Thank you again for your thoughtful comments and feedback. We sincerely hope that our response and revisions have addressed your concerns. If you feel that the updates satisfactorily clarify the issues you raised, we would be very grateful if you could consider updating your review to reflect your current assessment.
>
> If you still have any remaining concerns or questions, please do not hesitate to share them. We would be more than happy to provide further clarification.
>
> Thank you once again for your time and consideration.
>
> Sincerely,
>
> The Authors

---

> > ### Comment · Reviewer_cot2 · 2025-11-27
> >
> > Thank you for your response. I find it interesting that Reviewer `ukKu` intends to reproduce your results, and I am willing to wait for their findings.

---

### Official Review · Reviewer_vEHq · 2025-10-27

**Soundness:** 2
**Presentation:** 3
**Contribution:** 2
**Rating:** 4
**Confidence:** 3

**Summary:**

The paper introduces Stratified GRPO, whose key innovation is Stratified Advantage Normalization (SAN). SAN partitions trajectories into homogeneous strata based on structural properties and computes advantages locally within each stratum, ensuring fairer and more stable credit assignment. Experiments on multiple single-hop and multi-hop question answering benchmarks demonstrate consistent and significant improvements over standard GRPO (up to +11.3 points), as well as enhanced reward, stability, and policy quality.

**Strengths:**

1. Principled algorithmic design. The proposed Stratified Advantage Normalization (SAN) is simple, elegant, and theoretically grounded. The proofs of conditional unbiasedness and variance preservation are both rigorous and intuitive.
2. Empirical validation and robustness. Experiments are comprehensive and show consistent, significant improvements in both performance and training stability across diverse benchmarks.

**Weaknesses:**

1. Additive rather than integrative extensions. The “blended advantage” component, while useful in practice, feels more like an empirical stabilization trick rather than a deeply integrated conceptual innovation.
2. Lack of broader application tests. All experiments focus on QA-style search agents. It remains to be seen whether the proposed approach extends effectively to other multi-step or tool-augmented domains (e.g., code generation, planning, or dialogue systems).

**Questions:**

1. Have you tested Stratified GRPO in settings beyond search-based QA (e.g., tool-use agents, code synthesis, or reasoning benchmarks)?
2. Can you offer the hyperparameters for training, e.g. #rollouts? When trajectory distributions are highly imbalanced, how do you ensure stable variance estimates?

**Details Of Ethics Concerns:**

No.

---

> ### Author Response · Authors · 2025-11-22
> **Response 1**
>
> **Q1 The “blended advantage” component, while useful in practice, feels more like an empirical stabilization trick rather than a deeply integrated conceptual innovation.**
>
> We respectfully clarify that **Blended Advantage is not an additive heuristic**, but rather the **finite-sample realization necessary to operationalize SAN’s core theoretical contribution**.
>
> Our primary contribution is **SAN**. As shown in Theorem 4, SAN yields a _conditionally pure_ learning signal, exhibiting zero cross-stratum bias and unit within-stratum variance in the large-sample limit, thereby resolving the structural “cross-stratum bias” inherent in GRPO.
>
> However, RL training occurs in a finite-sample regime. When some strata are small, the empirical statistics $(\hat{\mu}_k, \hat{\sigma}_k)$ become noisy, and the “pure’’ SAN estimator might become unstable. Blended Advantage addresses exactly this issue by softly anchoring these high-variance local estimates to the more reliable global statistics.
>
> Table 2 supports this conceptual decomposition: **SAN alone already produces consistent and significant gains over GRPO**, validating the bias-reduction claim, while **adding Blended Advantage yields further improvements**, especially on multi-hop QA, demonstrating its role as a stability enhancement.
>
> Therefore, **Blended Advantage is not an ad-hoc workaround**, but a **principled stabilization mechanism required for SAN’s theoretical benefits to carry over to stochastic, finite-sample training**. We will revise Section 3 to make this role explicit.
>
> We appreciate the reviewer’s point and agree that this connection should be stated more clearly; we will strengthen the exposition accordingly.
>
> **Q2 Broader applicability of Stratified GRPO**
>
> We agree that broader validation is valuable. Our current experiments focus on factual QA, following Search-R1 [1]. However, **our algorithmic design is domain-agnostic**: the theory is derived for a general discrete stratum variable $S$ and does not assume QA-specific structure.
>
> To further evaluate generality, we add experiments on **tool-use agents in real-world multi-step settings**. Following VerlTool [2], we train a deep-research agent equipped with Google search engine and a sandboxed Python executor as tools, with Qwen3-8B as the base model. We evaluate the performance on the General AI Assistants benchmark (GAIA), which tests LLM assistants' comprehensive capabilities in real world tasks requiring reasoning, web browsing, and tool-use proficiency.
>
>
> | Method | Level 1 | Level 2 | Level 3 | Avg |
> |---|---:|---:|---:|---:|
> | **Reasoning without Tool** |  |  |  |  |
> | Qwen3-32B-thinking | 26.2 | 12.1 | 0 | 14.9 |
> | DeepSeek-R1-32B | 21.5 | 13.6 | 0.0 | 14.2 |
> | QwQ-32B | 30.9 | 6.5 | 5.2 | 18.9 |
> | GPT-4o | 23.1 | 15.4 | 8.3 | 17.5 |
> | DeepSeek-R1-671B | 40.5 | 21.2 | 5.2 | 25.2 |
> | **Tool Integrated Reasoning (Qwen3-8B)** |  |  |  |  |
> | Vanilla RAG | 28.2 | 15.4 | 16.7 | 20.4 |
> | Search-o1 | 35.9 | 15.4 | 0.0 | 21.4 |
> | WebThinker | 43.6 | 11.5 | 0.0 | 22.3 |
> | ReAct | 35.9 | 17.3 | 8.3 | 23.3 |
> | *RL-based Method* |  |  |  |  |
> | Qwen3-8B | 28.1 | 15.4 | 16.7 | 20.4 |
> | + GRPO | 48.7 | 32.7 | 16.7 | 36.9 |
> | + Stratified GRPO | **61.5** | **44.2** | **33.3** | **49.5** |
>
> Stratified GRPO substantially improves over GRPO (**+12.6 avg**) and outperforms other strong baselines. Importantly, these gains arise in a domain **far beyond QA retrieval over fixed corpora**, but involving open-web search, tool invocation decisions, longer-horizon planning, and reasoning. This directly demonstrates Stratified GRPO’s broader applicability to tool-augmented reasoning agents. We will include full experimental details and analysis in the revision.
>
>
>
> **Q3 Hyperparameters for training (e.g., #rollouts)**
>
> Appendix D.1 provides full training hyperparameters. In particular, we use 8 rollouts for experiments in the main paper, and 16 rollouts for the deep-research agent experiment in **Q2**.
>
> **Q4 When trajectory distributions are highly imbalanced, how do you ensure stable variance estimates?**
>
> This is an important point, and our method addresses it at both theory and implementation levels:
>
> 1. SAN guarantees zero-mean, unit-variance advantages within each stratum by construction (Theorem 4), and in the idealized limit SAN is globally unit-variance (Theorem 5). SAN also uses an $\epsilon$ stabilizer in the per-stratum normalization, preventing numerical blow-ups when $\sigma$ is small.
>
> 2. Blended advantage is explicitly designed for stablizing for small/imbalanced strata, as discussed in our response to **Q1**.
>
> 3. Empirically, these design choices yield stable training even in settings where GRPO collapses (Figure 1), supporting robustness under imbalance.
>
> [1] _Search-R1: Training LLMs to Reason and Leverage Search Engines with Reinforcement Learning._ COLM 2025.
> [2] _VerlTool: Towards Holistic Agentic Reinforcement Learning with Tool Use._ arXiv 2025.

---

> ### Author Response · Authors · 2025-11-26
>
> Dear Reviewer vEHq,
>
> Thank you again for your thoughtful comments and feedback. We sincerely hope that our response and revisions have addressed your concerns. If you feel that the updates satisfactorily clarify the issues you raised, we would be very grateful if you could consider updating your review to reflect your current assessment.
>
> If you still have any remaining concerns or questions, please do not hesitate to share them. We would be more than happy to provide further clarification.
>
> Thank you once again for your time and consideration.
>
> Sincerely,
>
> The Authors

---

### Official Review · Reviewer_DhK8 · 2025-11-03

**Soundness:** 4
**Presentation:** 4
**Contribution:** 3
**Rating:** 8
**Confidence:** 3

**Summary:**

The work proposes a variant of GRPO that is applicable when items in a batch are very heterogeneous, for example when RL training agentic behaviour.

The main idea is to partition each batch into similar trajectories, for example group several roll-out for the same prompt and with the same number of steps. They do, what I understand would be the typical computations for GRPO on that batch partition which they call a strata.

The benefit is that items within the strata are similar and thus the mean reward is more meaningful and thus the advantage is more meaningful. They correctly argue that in regular global normalization trajectories compete with others that might have fewer steps or different prompts and only those above the mean are reinforced which might fail to boost trajectories for harder examples.

The downside is that advantages are taken from a smaller sample and thus might be noisy, so they interpolate the stratified advantages with regular global advantages.

The authors prove a couple of fairly straightforward properties of their normalization method which formalizes the difference between the stratified and global normalizations.

The proposed approach is simple to implement, has no computation overhead, doesn't need too many hyperparameters and seems to do well empirically

**Strengths:**

Addresses a real problem that is becoming more evident as training switches to agents

Method is simple and sound, implementation is easy

Theoretical results look correct

Paper is well written, easy to read and doesn't overstate contributions

Nit: Good formatting for the result tables

**Weaknesses:**

Related work section was moved to Appendix. I feel this violates the length requirement.

**Questions:**

In Table 1: Are the large differences between Qwen-Base vs Qwen-instruct for NQ and Banboogle expected?

Improvements:

The core proposal is to partition the batches by some heuristic but the used heuristic is not mentioned in the experiments section. Please add.

In line 350 say "a linear combination" instead of "a convex combination"

---

> ### Author Response · Authors · 2025-11-22
> **Response 1**
>
> **Q1: Related work is in the Appendix.**
>
> Thank you for raising this. We initially placed the related work in the appendix to preserve space for the formal analysis and experiment results within the main-page limit, while still providing complete scholarly context. To avoid any concern about compliance or visibility, we will move the related work section back into the main text.
>
> **Q2: Are the large differences between Qwen-Base and Qwen-Instruct for NQ and Bamboogle expected?**
>
> Yes, these performance differences are expected. Our experiment setting follows Search-R1 [1], a pioneering work on LLM-based search agents. Notably, similarly large gaps between Qwen-Base and Qwen-Instruct on NQ and Bamboogle are also reported in Search-R1 (Table 2), which supports the consistency of our observations.
>
> **Q3: Add the stratification heuristic in the Experiments section.**
>
> Thank you for the suggestion. In all experiments, we stratify within each prompt by the trajectory’s search count/number of tool calls. This design is motivated by our analysis of heterogeneity drivers in Section 2. We will add a clarification in the Experimental Setup describing this stratification rule explicitly.
>
> **Q4: Wording change from “convex combination” to “linear combination.”**
>
> Thank you for the suggestion. We will correct the terminology in the updated PDF.
>
> [1] Search-R1: Training LLMs to Reason and Leverage Search Engines with Reinforcement Learning. COLM 2025.

---

> ### Comment · Reviewer_ukKu · 2025-11-26
> **You state "implementation is easy", but there is no code uploaded.**
>
> # As indicated in the title, I would like to respectfully inquire about the basis for your assessment of this paper. Specifically, you mentioned that “the implementation is easy,” yet no code has been provided. May I ask what leads you to that conclusion?

---

> > ### Author Response · Authors · 2025-11-26
> >
> > We thank **Reviewer ukKu** for engaging with the discussion.
> >
> > We respectfully disagree with the implication that the absence of a public code release prevents one from judging the implementation to be easy. The simplicity of our method follows from its algorithmic structure, not from the availability of our code. Concretely:
> >
> > -   **Our method is a minimal modification of GRPO.** Stratified GRPO uses exactly the same training pipeline as standard GRPO; the _only_ change is how advantages are computed, in which we partition trajectories by a simple strategy: the number of tool calls per prompt, as summarized in Algorithm 1. An existing GRPO implementation only needs a few extra lines to group trajectories.
> >
> > For example, in VERL’s official function for computing the GRPO advantage, rollouts in the batch are grouped into each prompt’s sampled rollouts based on the prompt index:
> >
> > ```
> > #VERL GRPO advantage
> > def compute_grpo_outcome_advantage(token_level_rewards: torch.Tensor,
> >                                    eos_mask: torch.Tensor,
> >                                    index: torch.Tensor,
> >                                    epsilon: float = 1e-6):
> >
> >     response_length = token_level_rewards.shape[-1]
> >     non_zero_mask = (token_level_rewards != 0)
> >     scores = (token_level_rewards * non_zero_mask).sum(dim=-1)
> >
> >     id2score = defaultdict(list)
> >     id2mean = {}
> >     id2std = {}
> >
> >     with torch.no_grad():
> >         bsz = scores.shape[0]
> >         for i in range(bsz):
> >             id2score[index[i]].append(scores[i])
> >
> >     # ... compute per-prompt mean/std and normalized advantages
> > ```
> >
> >
> > In our stratified version, the only modification needed is to extend the grouping key with the search-count information, i.e., replacing
> >
> > `id2score[index[i]].append(scores[i])`
> >
> > by
> >
> > `id2score[(int(index[i]), int(search_counts[i]))].append(scores[i])`
> >
> > while keeping the remaining identical. This is what we mean when we say that the implementation is easy.
> >
> > -   **The algorithm is fully specified in the paper.** Algorithm 1 gives a step-by-step description of Stratified GRPO, including the computation of global statistics, per-stratum statistics, SAN, and the blended advantage used in the policy gradient update. No new networks, optimizers, or training tricks beyond standard GRPO/RLVR are introduced, so an experienced practitioner can directly port these steps into any existing codebase.
> >
> > -   **Design choices and hyperparameters are completely documented.** Sections 3–5 and Appendix E detail the stratification strategy, model choices, and all training hyperparameters.
> >
> > -   **Code release plan.** In line with ICLR policy and standard practice, we are committed to releasing an implementation of Stratified GRPO upon acceptance, which will make the small set of required changes even more transparent to the community.
> >
> >
> > For these reasons, we believe it is accurate and reasonable for Reviewer DhK8 and for us to describe the implementation as “easy” relative to existing GRPO-based RLVR setups. We hope this clarifies the basis for that assessment for all reviewers and the area chairs.

---

### Author Response · Authors · 2025-11-26
**General Response**

We sincerely thank all reviewers for their thoughtful feedback and detailed suggestions. We have now updated the PDF to incorporate the requested experiments and textual changes. **The revisions are marked in blue in the updated PDF.**

Below we summarize the primary concerns and how they are addressed.

| # | Focus Area | Reviewer(s) | Revised Sections | Our Actions |
|---|------------|-------------|------------------|-------------|
| 1 | **Placement of Related Work** | DhK8 | Section 2: Related Work| We restored the Related Work section to the main paper. |
| 2 | **Specification of Stratification Heuristic & Wording Fix** | DhK8 | Line 339-340; Section 5.1: "Stratification Heuristic." | We now clearly describe the stratification heuristic used in experiments in Section 5.1. We also corrected the wording to say **“a linear combination”** instead of “a convex combination” at Lines 339-340 in the paper. |
| 3 | **Broader Applications Beyond Factual QA** | vEHq, cot2, ukKu | Section 5.2 "Results on Deep-Research Agent Tasks."; Table 2 | We add an experiment on **deep-research agent tasks** following VerlTool, using Qwen3-8B equipped with Google search engine and a sandboxed Python executor as tools. Evaluation is conducted on the General AI Assistants benchmark (GAIA), which tests LLM assistants' comprehensive capabilities in real world tasks requiring multi-step reasoning, web browsing, and tool-use proficiency. **Stratified GRPO improves substantially over GRPO across Level 1–3 tasks by 12.6 points on average**, and on the hardest Level-3 questions it achieves about a **2× relative improvement over GRPO (33.3 vs. 16.7)**|
| 4 | **Balancing Theory and Experiments** | cot2 | Section 3.3 "Structural Superiority over Global Normalization." | We streamlined the theoretical analysis in the main text and moved secondary derivations to the appendix. In parallel, we expanded the experiment section with more experiments, ablations, and discussions, yielding a better balance between theory and empirical validation. |
| 5 | **Sensitivity of Hyperparameter  $\alpha$** | cot2 | Section 5.3 "Robustness of Blending Coefficient $\alpha$."; Table 4 | We conducted a **sensitivity study over $\alpha$**. The results show that Stratified GRPO is effective and robust over a wide range of $\alpha$. |
| 6 | **Random-Strata Baseline** | cot2 | Section D "Validation of the Stratification Heuristic."; Table 7 | We add a sanity check for stratification strategy by testing a **random-strata baseline**, where trajectories are randomly assigned to strata. Results show that random stratification offers little or no improvement over vanilla GRPO, whereas our stratification provides consistent gains. This supports the claim that meaningful, structure-aware strata are crucial. |
| 7 | **Multiple Runs Evaluation** | ukKu | Section D "Multi-seed Stability Analysis. "; Table 6 | We report results over 3 random seeds in Table 6, providing means and standard deviations for both GRPO and Stratified GRPO. The multi-seed results show that our improvements are consistent and not due to isolated runs. |
| 8 | **Training Curves and Diagnostic Metrics** | ukKu |  Section D "Multi-seed Stability Analysis."; Figure 2; Figure 3 | We substantially expanded the diagnostics in Figure 2 and Figure 3. We include training curves and statistics such as **training reward, number of search calls, gradient norm, KL divergence, rollout lengths, policy entropy, global advantage variance after normalization, global advantage variance before normalization, and in-stratum advantage variance after normalization**. These are averaged over 3 runs. The diagnostics empirically support our theoretical claims and demonstrate the stability of Stratified GRPO. |

We believe these clarifications, additional experiments, and diagnostic analyses comprehensively address the reviewers’ concerns and significantly strengthen the paper’s empirical and conceptual story.

Best regards,

The Authors

---

### Meta-Review · Area_Chair_SLPn · 2026-01-11

**Summary:**

This paper studies reinforcement learning for LLM-based agents that rely on external tools such as search, where trajectories exhibit significant structural heterogeneity due to varying numbers and patterns of tool calls. The authors identify and formalize cross-stratum bias arising from the use of a single global baseline in standard policy gradient methods, which leads to distorted credit assignment across heterogeneous trajectories. To address this issue, the paper proposes Stratified GRPO, introducing Stratified Advantage Normalization (SAN) to compute advantages within structurally homogeneous trajectory groups. Theoretical analysis claims that SAN removes cross-stratum bias while preserving unbiasedness and variance control, and empirical results across multiple search-heavy benchmarks show consistent and substantial performance gains over standard GRPO.

**Reviewer Concerns:**

- Limited depth and novelty of the theoretical analysis: Reviewers felt that the theoretical results are largely straightforward and, in some cases, overly emphasized relative to their technical substance. Several results are seen as either trivial, well-known, or not unique to the proposed method, with limited new conceptual insight.

- Imbalance between theory and experiments: The paper devotes substantial space to theoretical analysis, while the experimental section is comparatively underdeveloped. Reviewers suggested a better balance, with more empirical evidence to justify and validate the theoretical claims.

- Insufficient empirical validation of key claims: Core theoretical assertions—such as variance reduction and improved training stability—are not directly supported by targeted experimental diagnostics (e.g., gradient norm stability, variance of advantage estimates, KL divergence trajectories).

- Limited experimental scope and generality: All evaluations focus on search-based question answering tasks. The applicability of the proposed method to other tool-augmented or multi-step settings (e.g., code generation, planning, or multi-tool agents) remains untested.

- Limited analysis of hyperparameter sensitivity: Important hyperparameters (e.g., blending weights, stratification thresholds) are not systematically studied. Sensitivity analyses beyond a small number of settings are needed to understand robustness.

- Additive rather than integrated extensions: Some components (e.g., blended advantages) are perceived as empirical stabilization heuristics rather than deeply integrated conceptual contributions.

**Reviewer Scores:**

No, except Reviewer DhK8, everyone else is concerned about the theoretical novelty and reproducibility of the empirical results. They all kept their scores unchanged.

---

### Decision · Program_Chairs · 2026-01-26

Reject